# Visual information is broadcast among cortical areas in discrete channels

Yiyi Yu[1], Jeffery N Stirman[1†], Christopher R Dorsett[1‡], Spencer LaVere Smith[1,2]*

[1]Department of Electrical and Computer Engineering, University of California, Santa Barbara, Santa Barbara, United States; [2]Dynamical Neurosciences, University of California, Santa Barbara, Santa Barbara, United States

## eLife assessment

.This **important** study uses state-of-the-art, multi-region two-photon calcium imaging to characterize the statistics of functional connectivity between visual cortical neurons. Although alternative interpretations may partially account for the data, the study provides **solid** evidence that functionally distinct classes of neurons convey visual information via parallel channels within and across both primary and higher-order cortical areas.

*For correspondence:
sls@ucsb.edu

Present address: †LifeCanvas, Cambridge, United States; ‡SanDiego, San Diego, United States

**Abstract** Among brain areas, axonal projections carry channels of information that can be mixed to varying degrees. Here, we assess the rules for the network consisting of the primary visual cortex and higher visual areas (V1-HVA) in mice. We use large field-of-view two-photon calcium imaging to measure correlated variability (i.e. noise correlations, NCs) among thousands of neurons, forming over a million unique pairs, distributed across multiple cortical areas simultaneously. The amplitude of NCs is proportional to functional connectivity in the network, and we find that they are robust, reproducible statistical measures and are remarkably similar across stimuli, thus providing effective constraints to network models. We used these NCs to measure the statistics of functional connectivity among tuning classes of neurons in V1 and HVAs. Using a data-driven clustering approach, we identify approximately 60 distinct tuning classes found in V1 and HVAs. We find that NCs are higher between neurons from the same tuning class, both within and across cortical areas. Thus, in the V1-HVA network, mixing of channels is avoided. Instead, distinct channels of visual information are broadcast within and across cortical areas, at both the micron and millimeter length scales. This principle for the functional organization and correlation structure at the individual neuron level across multiple cortical areas can inform and constrain computational theories of neocortical networks.

## Introduction

Neurons have characteristic preferences or tuning, which are variables (e.g. stimulus or behavior variables) that correlate with their spiking activity. Neuronal spiking activity is transmitted via axonal projections to other brain areas. In the early stages of visual processing, visual information can be preserved. For example, the retina-to-lateral geniculate nucleus (LGN) network tends to preserve unmixed channels by ensuring that axons from retinal ganglion cells with similar tuning converge on individual LGN neurons (*Liang et al., 2018*). By contrast, the LGN-to-primary visual cortex (V1) network famously mixes channels to give neurons receptive fields with both dark-sensing and light-sensing subregions, and thus robust orientation tuning (*Hubel and Wiesel, 1962*). That said, discrete visual information can be transmitted from the retina to the cortex through non-mixing channels. For example, when direction-selective neurons in the retina are genetically ablated, there is a decrease in direction-selective neurons in cortex (*Rasmussen et al., 2020*).

In the visual cortical system in mice, the primary visual cortex (V1) and its projections to multiple higher visual areas (HVAs) span millimeters (*Wang and Burkhalter, 2007*). Local networks within V1 can have precise local (<50 μm) cellular-resolution functional connectivity (*Ko et al., 2011*). Studies of longer-range, millimeter-scale networks typically lack cellular resolution, but there are general biases observed. Neurons in V1 and HVAs respond to diverse visual stimuli (*Yu et al., 2022*; *de Vries et al., 2020*) and are sensitive to a broad range of features including orientation and spatiotemporal frequencies (*Andermann et al., 2011*; *Marshel et al., 2011*). Although individual V1 neurons broadcast axonal projections to multiple HVAs (*Han et al., 2018*), the spatiotemporal frequency preferences of these feedforward projections generally match those of the target HVAs (*Glickfeld et al., 2013*; *Han and Bonin, 2023*; *Kim et al., 2018*). Feedback connections from HVAs carry frequency-tuned visual signals as well (*Huh et al., 2018*). Thus, there are HVA-specific spatiotemporal biases, but cellular resolution and millimeter-scale principles for cortical wiring remain to be elucidated.

In the current study, we investigated the degree of channel mixing between distinctly tuned neurons in the V1-HVA networks of mice by measuring the noise correlations (NC, also called spike count correlations *Vinci et al., 2016*) between functional tuning classes of neurons. Functional tuning classes were defined using an unbiased clustering approach (*Han et al., 2022*; *Yu et al., 2022*; *Baden et al., 2016*). Large field-of-view (FOV) calcium imaging enabled us to densely sample across millimeters of cortical space, simultaneously observing large and dense samples of neurons in these tuning classes within and across cortical areas (*Yu et al., 2021*; *Stirman et al., 2016*). NCs are due to connectivity (direct or indirect connectivity between the neurons, and/or shared input), and thus provide a trace of connectivity (*Cohen and Kohn, 2011*; *Vinci et al., 2016*; *Snyder et al., 2015*). In particular, the connectivity that underlies NCs is effective in vivo, during normal sensory processing. Thus, NCs provide a complement to purely anatomical measures of connectivity. In fact, activity-based estimates of neuronal networks can provide higher fidelity measures than anatomy-based studies (*Randi et al., 2023*). We find that NCs are a reliable measure at the population level. We also find that neuron classes can be categorized into six functional groups, and NCs are higher within these groups (and even higher within classes), both within and across cortical areas, indicating unmixed channels in the network preserve information. Moreover, we find that naturalistic videos draw upon the same functional networks, and modeling suggests that recurrent connectivity rather than bottom-up or top-down input is critical for stabilizing these networks.

## Results

### Visual cortical neurons form six tuning groups

To measure neuronal activity, we used multi-region population calcium imaging of L2/3 neurons in V1 and four HVAs (lateromedial, LM; laterointermediate, LI; anterolateral, AL; and posteromedial, PM) using a multiplexing, large field-of-view two-photon microscope with subcellular resolution developed in-house (*Stirman et al., 2016*; *Figure 1A*). Mice expressed the genetically encoded calcium indicator GCaMP6s (*Madisen et al., 2015*; *Chen et al., 2013*) in cortical neurons. We located the V1 and HVAs of each mouse using retinotopic maps obtained by intrinsic signal optical imaging (*Marshel et al., 2011*; *Smith et al., 2017*; *Figure 1—figure supplement 1*). We imaged neurons in two to four cortical areas simultaneously (*Figure 1A*), while mice viewed stimuli on a video display. We typically imaged neurons in V1 and one or more HVAs. Up to 400 neurons (V1: 129±92; HVAs: 94±72; mean ± SD; only counting reliably responsive neurons used in subsequent analysis, see Materials and methods) were recorded per imaging region (500x500 μm²). The imaging regions were matched for retinotopy so that the neurons in the simultaneously imaged areas had overlapping receptive fields (RFs). Calcium signals were used to infer probable spike trains for each neuron, as our previous study (*Yu et al., 2022*). We mapped RFs for individual neurons and populations using small patches of drifting gratings (*Figure 1—figure supplement 1B*, C). Neurons in HVAs (LM, AL, PM, and LI) had significantly larger RFs than V1 neurons (*Figure 1—figure supplement 1D*). Population RFs for a 500x500 μm² imaging region of HVAs covered significantly larger portions of visual space than that of V1 (*Figure 1—figure supplement 1D*), as expected given their differing magnification factors (*Wang and Burkhalter, 2007*; *Smith et al., 2017*). The overlap of population RFs confirmed that simultaneously imaged cortical areas (V1 and HVAs), each containing ~100 neurons, responded to stimuli in the same region of the

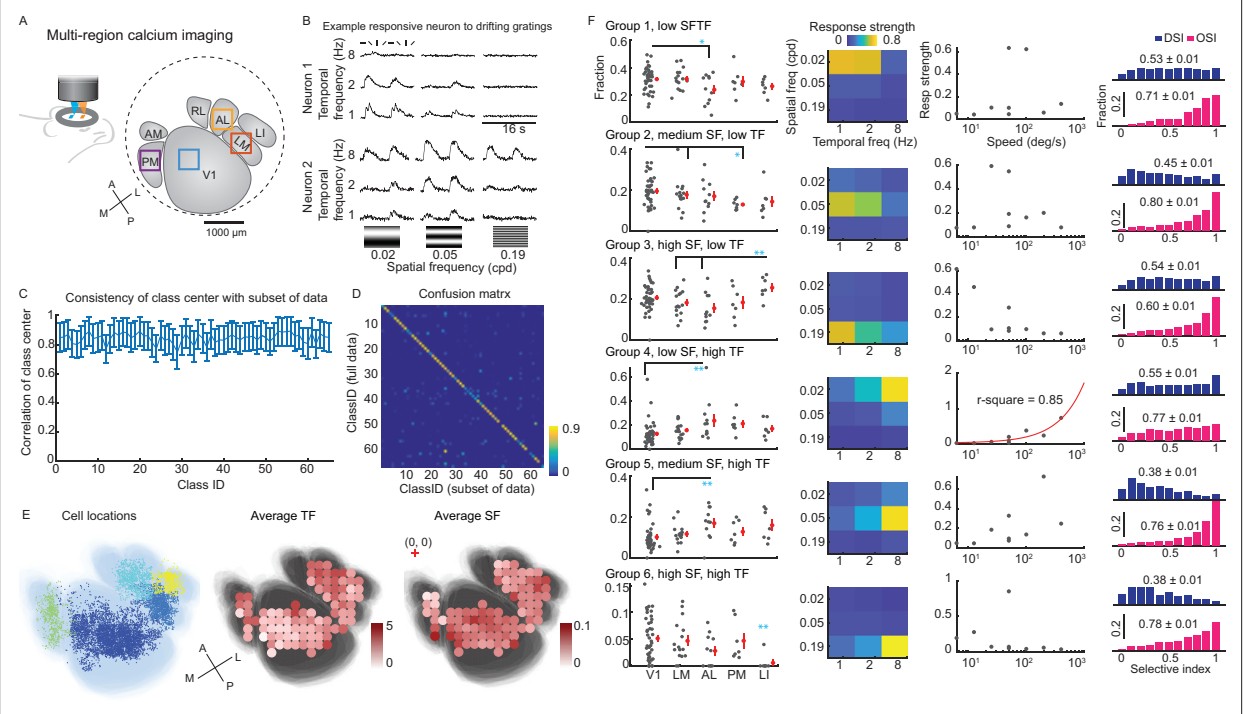

**Figure 1.** Functional groups of mouse visual neurons. (**A**) Diagram of multi-region two-photon imaging of mouse V1 and HVAs, using a custom wide field-of-view microscope. Example imaging session of the simultaneous recording session of V1, LM, AL, and PM. Squares indicate 500 µm wide imaging regions. (**B**) Example responses from two neurons (mean calcium trace) to drifting gratings with eight directions at various SF-TF frequencies. (**C**) Neurons were distributed into 65 different classes using GMM (*Figure 1—figure supplements 1–2*). The mean correlation coefficients of the center of each class (in principal component space) between GMMs of 10 permutations of a random subset of neurons. (**D**) The confusion matrix shows that individual neurons are likely (>90%) to remain in the same class even when only a random subset of neurons is used to train the GMM (horizontal), compared to the full data set (vertical). (**E**) Center of individual neurons (left) overlay on an average visual cortex map. The average visual cortex map was generated by affine registration of visual area maps from all experiments. Neurons are colored by visual areas. Middle, average preferred TF exhibits spatial dependency over the visual cortex (TF: $A \rightarrow P$, $cor = -0.25$, $p = 0.015$, $M \rightarrow L$, $cor = 0.36$, $p = 0.0004$). Right, the average preferred SF (right) exhibits spatial dependency over the visual cortex (SF: $A \rightarrow P$, $cor = 0.35$, $p = 0.0005$, $M \rightarrow L$, $cor = -0.06$, $p = 0.54$). Colored dots indicate the average TF and SF (computed with >30 neurons) within patches (180 µm x 180 µm local areas), overlaid on a map of V1 and HVAs. (**F**) These 65 classes were manually arranged into six tuning groups based on spatial frequency and temporal frequency (SF-TF) tuning preferences. Column 1, the fraction of neurons in different SF-TF groups. Dots represent individual sessions. Statistical significance was tested by the Ranksum test (*, p<0.05, **, p<0.01). Column 2, the characteristic SF-TF responses of each tuning group. Column 3, speed tuning of tuning groups. Column 4, distribution of cells' orientation selectivity index (OSI) and direction selectivity index (DSI). The number of neurons belonged to the six tuning groups combined: V1, 5373; LM, 1316; AL, 656; PM, 491; LI, 334 (refer to Materials and methods for neuron selection). These six groups provide a compact way of summarizing response diversity, but as shown later, the granularity of the 65 classes provides a superior match to the network properties (*Figure 4F*).

The online version of this article includes the following figure supplement(s) for figure 1:

**Figure supplement 1.** Functional groups by multi-region two-photon calcium imaging.

**Figure supplement 2.** t-SNE embedding of GMM classes.

**Figure supplement 3.** Spatial modulation on SF-TF and orientation tuning.

screen (*Figure 1—figure supplement 1C*). These experiments were repeated in 24 mice for a total of 17,990 neurons and NCs were measured for a total of 1,037,701 neuron pairs (*Supplementary file 1*).

Mouse V1 and HVA neurons exhibit diverse tuning preferences to drifting grating stimuli, in terms of spatiotemporal preferences and sharpness of orientation and direction tuning (*Marshel et al., 2011*; *Andermann et al., 2011*; *de Vries et al., 2020*). Previous studies suggested that the axonal projections from V1 to HVAs match the spatiotemporal preferences of the target HVAs (*Glickfeld et al., 2013*). We sought to determine whether this was a general principle that extended across V1 and HVAs. We recorded neuronal responses from V1 and multiple HVAs (LM, LI, AL, and PM) to sine-wave drifting grating stimuli with various spatiotemporal properties (8 directions x 3 spatial frequencies x 3 temporal frequencies for a total of 72 conditions; *Figure 1B*). HVAs exhibited similar

responsiveness and reliability to the 72 different parameterized drifting gratings. V1 and LM were only marginally more reliable than other areas (*Figure 1—figure supplement 1E*).

To obtain a granular, data-driven way to classify neurons, neuronal responses were partitioned into 65 tuning classes using an unbiased Gaussian Mixture Model (GMM; *Figure 1—figure supplement 1F*, *Figure 1—figure supplement 2*). This GMM classification was reliable, in that the center of the Gaussian profile of each class was consistent among GMMs of random subsets of neurons (*Figure 1C*). Neurons were consistently classified into the same class (*Materials and methods*; *Figure 1D*).

To examine the spatiotemporal frequency selectivity of HVAs, we manually partitioned the 65 GMM classes into six spatial frequency (SF) - temporal frequency (TF) selective groups (*Figure 1F*). Groups 1, 2, and 3 all prefer low TF (1–2 Hz) and prefer low SF (0.02 cpd), medium SF (0.05 cpd), and high SF (0.19 cpd), respectively. Groups 4, 5, and 6 all prefer high TF (8 Hz) and prefer low SF, medium SF, and high SF, respectively. Group 4 (low SF, high TF) was the only group that exhibited increasing responses to the drift speed of the grating stimulus (drift speed = TF/SF, and is measured in deg/s). These groupings were robust and reliable (*Figure 1—figure supplement 1G and H*). Similar to the previous report, V1 and HVAs have overlapping SF and TF selectivity (*Marshel et al., 2011*), with a trend of larger preferred TF from the posterior-medial to the anterior-lateral visual cortex, and a trend of increasing preferred SF from the anterior to the posterior visual cortex (*Figure 1E*). Specifically, AL had a larger fraction of neurons tuned to low SF high TF (Group 4, speed tuning group), and a lower fraction of neurons tuned to low SF-TF (Group 1), compared to V1 (*Figure 1F*). PM had a lower fraction of low TF medium SF neurons compared to V1 and LM (*Figure 1F*). LI had a larger fraction of neurons tuned to high SF and low TF than AL and LM (Groups 3) and had a lower fraction of neurons tuned to high TF and high SF (Group 6; *Figure 1F*).

Neurons in all six groups exhibited orientation and direction selectivity (*Figure 1F*). The preferred directions of neurons were evenly distributed in V1 and HVAs, except high SF groups (Group 3 and 6) of AL, PM, and LI biased to cardinal directions (*Figure 1—figure supplement 3A*). The unbiased GMM approach revealed that the orientation selectivity index (OSI) and direction selectivity index (DSI) of visual neurons were jointly modulated by SF and TF (*Figure 1—figure supplement 3B*). Neurons tuned to high SF and low TF (Group 3) exhibited lower OSI in all tested areas than all of the other groups (Group 3: mean OSI = 0.6; other groups ranged from 0.71 to 0.80; p<0.0001, one-way ANOVA with Bonferroni correction; *Figure 1F*, *Figure 1—figure supplement 3B*). Neurons tuned to high TF and medium-high SF (Groups 5 and 6) exhibited lower direction selectivity than other groups (Group 5, 6, mean DSI 0.38; other groups, mean DSI 0.45–0.54; p<0.0001, one-way ANOVA with Bonferroni correction; *Figure 1F*, *Figure 1—figure supplement 3B*).

In summary, we found that neurons in V1 and HVAs are jointly selective to the spatiotemporal frequency and the drifting orientation/direction of gratings. Consistent with (*Marshel et al., 2011*), V1 and LI have higher average preferred SF compared to AL (p<0.0001, one-way ANOVA with Bonferroni multi-comparison), while V1 has a lower average preferred TF than all tested HVAs (*Figure 1—figure supplement 3C*). In contrast to *Marshel et al., 2011*, we found AL has a higher average preferred TF than other visual areas, including LM (p<0.0001, one-way ANOVA with Bonferroni multi-comparison; *Figure 1—figure supplement 3C*). The discrepancy may be explained by the joint selectivity of spatio-temporal frequency and the orientation/direction, and the different stimuli used in these studies.

## NCs are robust measurements of functional networks

A unique aspect of this data set is the scale of the NC measurements, which allows us to measure NCs with individual neuron precision within dense local networks and over millimeter-length scales, in awake mice. Pioneering work in this area focused on local populations, typically less than 1 mm across (*Ko et al., 2011*; *Harris and Mrsic-Flogel, 2013*; *Lee et al., 2016*; *Wertz et al., 2015*) or electrode studies over long distances with few neurons in each location (*Clay Reid and Alonso, 1995*; *Siegle et al., 2021*). To investigate the V1-HVA functional network, we computed the NCs of pairs of neurons within individual cortical areas (within-area NC), and NCs for pairs of neurons where the two neurons are in different cortical areas (inter-area NC; *Figure 2A*). NCs are computed from the residual activity of individual neurons after subtracting the expected neuron firing on nominally identical trials. In this section, we evaluated the fidelity of our NC measurements. We also considered potential bias or noise due to the imprecision of spike inference and the finite number of trials.

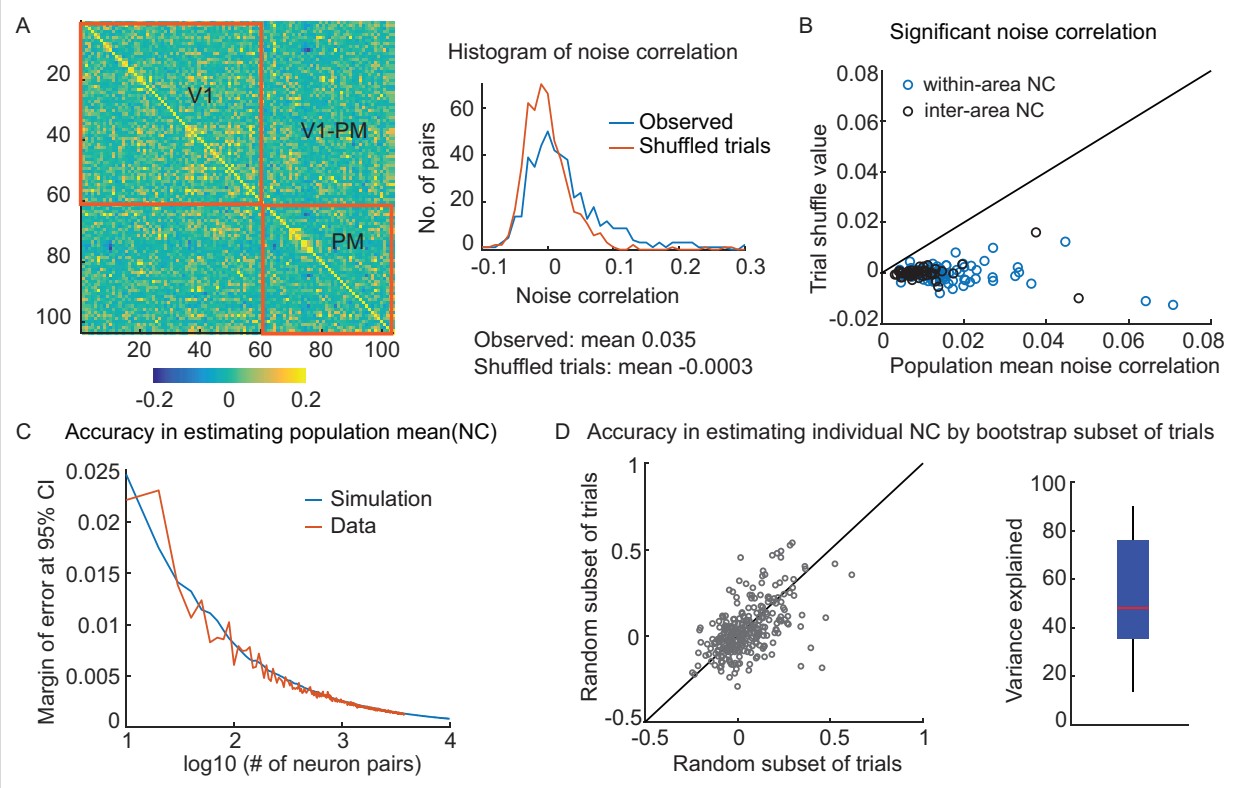

**Figure 2.** Noise correlation measurements are reliable. (**A**) (*left*) An example covariance matrix of a simultaneous recording of neuronal activity in V1 and PM. (*right*) NC histogram of an imaging session (blue, note the large positive tail), compared to control, trial-shuffled data (red). (**B**) The observed population mean NC is always larger than control values (the NC after trial shuffling). Circles indicate the value of individual experiments. (**C**) Population NCs are precise measures. The margin of error at a 95% confidence interval (CI) of the population mean NC reduces rapidly with an increasing number of neuron pairs (72-time bins x 10 trials). With the experimental size of the population (>100 neuron pairs), the estimation precision surpasses the 0.01 level. (**D**) (*left*) The NC of individual neuron pairs can be computed using different random subsets of trials, yet reliably converges on similar values (*right*) The variance of NC computed using a subset of trials is explained by the variance in the held-out subset of trials (53 ± 24 % variance explained; total 204 populations). Each subset contains half of all the trials. The variance explained is defined as the $R^2$ of the linear model.

The online version of this article includes the following figure supplement(s) for figure 2:

**Figure supplement 1.** Spatial modulation on SF-TF and orientation tuning.

We first evaluated the accuracy of NC calculations using inferred spikes from calcium imaging. We characterized the accuracy of spike inference using previously published data of simultaneous two-photon imaging and electrophysiological recording of GCaMP6s-positive neurons from mouse V1 (*Chen et al., 2013*). Consistent with a previous benchmark study on spike train inference accuracy (*Theis et al., 2016*), we found that the spike train inference methods used in the current study recovered 40–70% of the ground truth spikes (*Figure 2—figure supplement 1A*). We found that a similar fraction of spikes was missing regardless of the inter-spike interval. Nevertheless, the inferred spike train was highly correlated with the true spike train (*Figure 2—figure supplement 1A*; linear correlation, *r*=0.80 ± 0.03 [n=6]). Computing correlations between pairs of neurons using their inferred spike trains accurately reproduced the true correlation values (*Figure 2—figure supplement 1B*; linear correlation, *r*=0.7). We further examined the fidelity of correlation calculations using modified spike trains that are missing spikes. We examined randomly deleting spikes, deleting isolated spikes, or deleting spikes within bursts (*Figure 2—figure supplement 1D*; Materials and methods). We found that at the 1 s time scale, correlation calculations were tolerant to these spike train perturbations. The fidelity of correlation computations was >0.6 with up to 60% missing spikes (*Figure 2—figure supplement 1E*; Materials and methods). Thus, with conventional spike inference accuracy, about 80% variance of the true correlation is recovered (*Figure 2—figure supplement 1E*). Thus, NCs are a robust measurement even with imperfectly inferred spike trains.

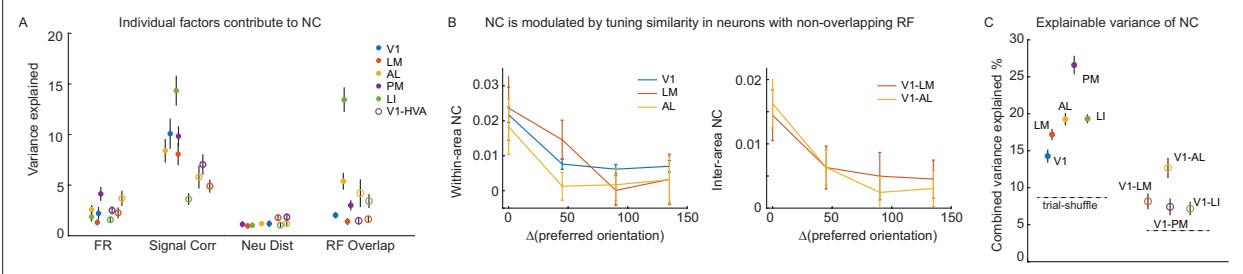

**Figure 3.** Factors that contribute to mesoscale NC. (**A**) The variance of within-area and inter-area NCs (during grating stimuli) is explained by individual factors including firing rate (FR), signal correlation (signal corr.), neuron distance (neu. dist.), and receptive field (RF) overlap. (**B**) NCs of neurons with non-overlapping RF are modulated by orientation tuning similarity (within-area, $P_{V1} < 10^{-4}$ (N=3401), $P_{LM} = 0.03$ (N=181), $P_{AL} = 0.019$ (N=284); inter-area: $P_{V1-LM} = 0.019$ (N=650), $P_{V1-AL} = 0.0004$ (N=998); t-test). (**C**) The variances of within-area NCs and inter-area NCs are explained by FR, signal corr., neu. dist., and RF overlap combined. The variance explained is a multi-linear regression model's $R^2$. (**A, C**) The error bar indicates the standard error of the mean of permutations. A subset of 100 neuron pairs was randomly selected for each permutation.

The online version of this article includes the following figure supplement(s) for figure 3:

**Figure supplement 1.** Factors contribute to the variance of NCs.

**Figure supplement 2.** Distance dependence of inter-area NC explained by retinotopic map.

Next, we evaluated the robustness of NC measurements, given the finite number of trials that are feasible to obtain. We computed NCs for both within-area and inter-area neuron pairs (*Figure 2A*). NCs were computed using spike counts within 1 s bins, similar to previous work with electrophysiology (*Cohen and Kohn, 2011*; *Smith and Sommer, 2013*). Although both within-area and inter-area NCs had wide distributions (range: –0.2–0.6), the mean NCs across a population were positive and at least five times larger than control data, which are NCs computed after shuffling the trials (5–20-fold, 25–75% quantile; *Figure 2B*). The estimation of the population mean NC converges fast with increasing numbers of neurons, as suggested by both simulation and experimental data (e.g. the margin of error at 95% CI for mean NC is 0.008 for 100 neuron pairs, *Figure 2C*). While the population-level NC calculations are reliable, the NC estimation of individual neuron pairs is noisier due to the limited number of trials, albeit positively correlated (*Figure 2D*). A linear model explains about 53 ± 24% of the variance between NCs for individual neuron pairs computed using different random subsets of trials. In summary, this evidence indicates that NCs can be accurately measured at the population level with our large FOV calcium imaging methods, despite imperfect spike train inference and a finite number of trials.

## Tuning similarity is a major factor in the V1-HVA functional network

Having established that NC measurements are reliable and robust at the population level, we examined potential NC-regulating factors, including firing rate (joint across the pair), the physical distance between the neurons (laterally, across the cortex), signal correlation (SC, the similarity between two neurons' average responses to stimuli), and RF overlap. We assessed the contributions of individual factors using a linear model. We found that both within- and inter-area NCs are similarly modulated by the aforementioned factors (*Figure 3A*; $r^2$ of the linear regression model). SC is the most pronounced factor that explains about 10% of the variance of within-area NCs, and about 5% of the variance of inter-area NCs (*Figure 3—figure supplement 1A*). The fraction of RF overlap contributes about 6% and 3% variance of within- and inter-area NCs, respectively (*Figure 3—figure supplement 1B*). The firing rate explained about 2% of the variance of within- and inter-area NCs (*Figure 3—figure supplement 1C*). The cortical distance explained about 2–3% of the variance of the NCs (*Figure 3—figure supplement 2*).

SC, RF overlap, and firing rate positively regulate both within- and inter-area NC, with SC providing the strongest predictor. Cortical distance negatively regulates the within-area NC (*Figure 3—figure supplement 2A*), as expected. However, cortical distance is non-monotonically related to the inter-area NC (*Figure 3—figure supplement 2B*). This can be explained by how retinotopic organization progresses across area boundaries. That is, when the RF locations are accounted for, a monotonic decrease in NC with distance can be recovered (*Figure 3—figure supplement 2C–H*).

We then evaluated whether NCs are modulated by tuning similarity independent of RF overlap. In the subset of orientation-selective neurons, both within- and inter-area NCs were significantly modulated by orientation-tuning selectivity. That is, neuron pairs that shared the same preferred orientation exhibited higher NCs (*Figure 3—figure supplement 1D*). NCs of a subset of neurons with non-overlapping RFs were significantly higher when the neurons shared the same preferred orientation (*Figure 3B*; t-test, $p < 0.05$ for V1, LM, and AL neuron pairs, insufficient data for PM and LI). This result confirms that the connectivity between neurons is modulated by tuning similarity (SC) independent of RF overlap, over millimeter distance scales.

Overall, about 20% of the variance of within-area NCs, and 10% of the variance of inter-area NCs are explained by the aforementioned factors jointly (*Figure 3C*; $r^2$ of multi-linear regression model). Although inter-area NCs have a smaller mean and variance, they are less predictable by known factors (within-area NCs pooled over all tested area $0.012 \pm 0.052$, inter-area NCs between V1 and all tested HVAs $0.0063 \pm 0.04$; both t-test and F-test $p < 10^{-4}$). In an expansion of prior work on local functional sub-networks (*Lee et al., 2016*; *Wertz et al., 2015*; *Ko et al., 2011*; *Harris and Mrsic-Flogel, 2013*), we find that signal correlation is the strongest factor regulating both within-area and inter-area NC networks, suggesting that neurons exhibiting similar tuning properties are more likely to form functional sub-networks across a broad spatial scale, spanning millimeters in the mouse V1-HVA network.

## Neurons are connected through functionally distinct, unmixed channels

Since HVAs exhibit biased SF-TF selectivity (*Figure 1E*), and tuning selectivity (i.e. SC) is a major factor for functional connectivity even across the millimeter length scale (*Figure 3A*), we assessed the precision of this network in the tuning (SC) dimension. We performed additional analysis to determine whether the ST-TF biases in HVAs could be due to simple, weak biases in the NC connectivity. Alternatively, there could be non-mixing channels of connections in the V1-HVA network to preserve information among similarly tuned neurons. We found evidence for this latter situation. Moreover, we found that the non-mixing channels consist of a greater number of neuron pairs with high NCs.

For this analysis, we focused on neuron pairs with high NCs, which we defined as NCs >2.5 standard deviations above the control (trial-shuffled) NCs for the population (*Figure 4A*). We focused on these high NC pairs because they can represent high-fidelity communication channels between neurons. Within each SF-TF group, for both V1 and HVAs, about 10–20% of neuron pairs exhibited high NCs, in contrast to 5% for inter SF-TF group connections (*Figure 4A and B*). The fraction of pairs that exhibit high NCs is relatively uniform across tuning groups and HVAs with a few exceptions (*Figure 4B*). For example, in HVA PM group 3 contains a higher fraction of high-fidelity connections than the other HVAs. Overall, these results show that mixing between groups is limited, and instead, group-specific high-NC sub-networks exist between neurons across millimeters of cortical space.

Prior findings from studies of axonal projections from V1 to HVAs indicated that the number of SF-TF-specific boutons underlies the biased frequency tuning in LM, AL, and PM, while the strength of a small fraction of speed tuning boutons contributes to the biases in speed tuning among these HVAs (*Glickfeld et al., 2013*). Although the functional connectivity is not completely defined by the feedforward axonal projections from V1 (*Huh et al., 2018*), this number vs. strength question is one that we can address with our data set. We found that the biased representation of SF-TF among HVAs is linearly related to the number of neuron pairs with high NCs (*Figure 4C*). By contrast, the average NC value (mean strength of connection) for each tuning group is not linearly related to the fraction of SF-TF groups in each HVA (*Figure 4D*). That is, the biasedly represented tuning group in HVA does not tend to have a higher functional connectivity with V1. Thus, the biases in SF-TF representation are likely related to the abundance of significant SF-TF-specific connections, but not the strength of the connections.

To this point, we have focused on the six SF-TF groups. The evidence supports group-specific channels among these neurons. However, these six groups originated with 65 *classes* from data-driven GMM clustering and were manually collected into the six SF-TF *groups* (*Figure 1*). The trends we see for groups may reflect general SF-TF biases. In that case, we would expect that the in-class NCs would exhibit similar distributions of NCs as the in-group NCs. However, there might be further precision in the specific channels not captured by the SF-TF groups. A hint towards that can be seen in the fact that orientation tuning can modulate NCs (*Figure 3*). Indeed, when we plotted the NC distribution for in-class neuron pairs and compared it to the distribution for in-group neuron pairs, we found a

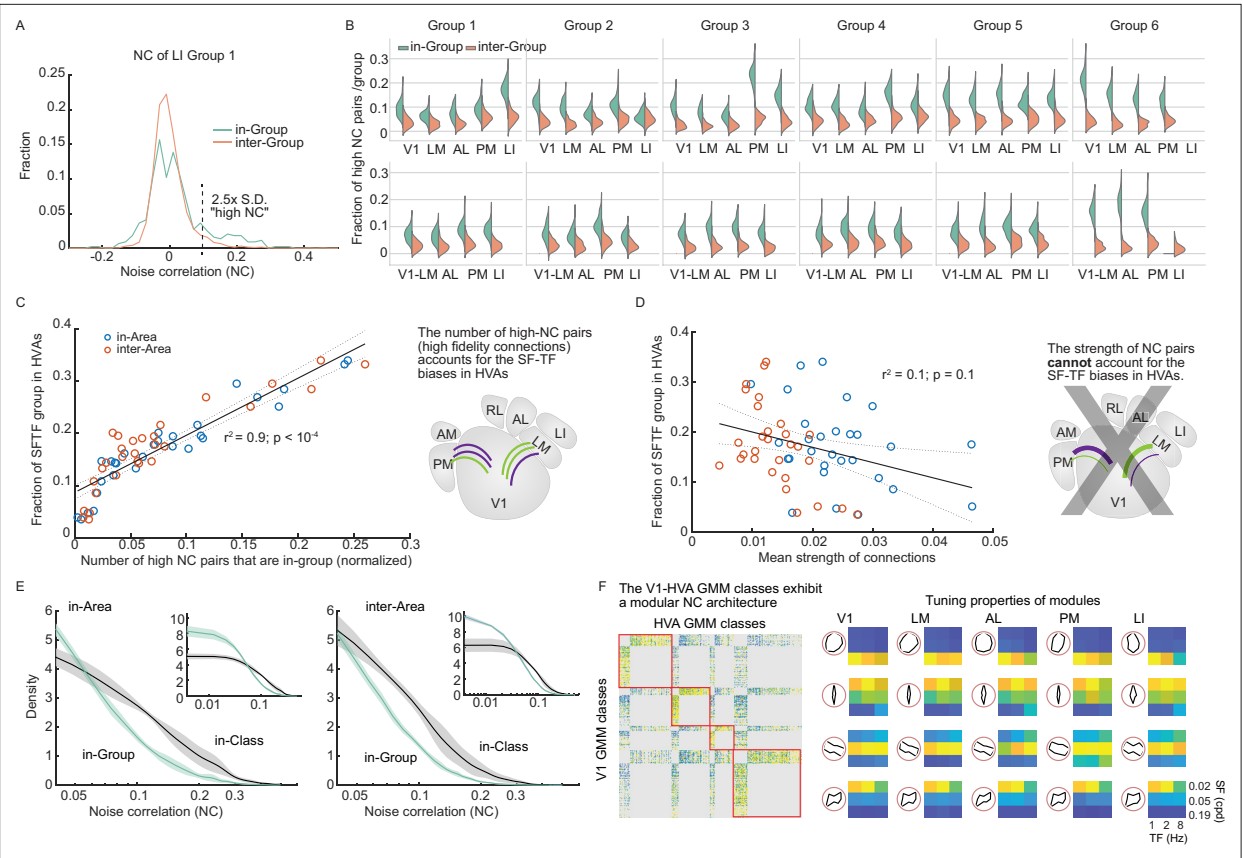

**Figure 4.** High-fidelity tuning-specific V1-HVA communication channels. (**A**) The distribution of NCs of a subset of LI neurons from tuning group 1 (blue) and NCs of a subset of LI neurons between group 1 and other tuning groups (red). In the next panel, we will focus on the positive tail: the portion of the distribution that is over 2.5 x of the S.D. (**B**) Fraction of neuron pairs with high NCs (>2.5*S.D. of trial-shuffled NCs) for within-group and inter-group pairs. Neurons within a group have a larger fraction of neuron pairs exhibiting high-fidelity connections (all comparisons, t-test, p<0.0001). Distributions were generated with 100 permutations. (**C**) The fraction of high-fidelity connections is linearly related to the fraction of SF-TF groups in each HVA ($r^2$=0.9; p<0.0001). The X-axis indicates the number of high NC neuron pairs in each group in the simultaneously imaged V1-HVA populations divided by the total number of high NC pairs in the V1-HVA population imaged (sum over all groups). (*right*) This result is summarized in a diagram indicating that area-specific SF-TF biases correlate with the number of high-fidelity functional connections. (**D**) The average NC value (mean strength of connection) for each tuning group is not linearly related to the fraction of SF-TF groups in each HVA ($r^2$=0.1; p=0.1). (*right*) This result negates the hypothesis suggested by the diagram, where area-specific SF-TF biases correlate with the strength of functional connections. Instead, the number of connections (panel **C**) seems to account for the observed trends. (**E**) Density function plots of NCs for in-area (left) or inter-area (right) neuron pairs that shared the same GMM-based class (65 classes) or group (six SF-TF preference groups) indicate that the more granular, GMM-based class categorization accounts for the structure of the NC network with higher fidelity than the coarser SF-TF groups (full scale is inset, bin size is 0.00875). (**F**) The functional connectivity matrix for the V1-HVA network between GMM classes exhibits a modular structure. (Right) Each module has a particular tuning selectivity and SF-TF bias. The value of this functional connectivity matrix was the fraction of high NC pairs.

The online version of this article includes the following figure supplement(s) for figure 4:

**Figure supplement 1.** Connectivity between GMM classes.

pronounced positive tail for the in-class distribution (***Figure 4E***). Thus, the GMM classes provide relevant, granular labels for neurons, which form functional sub-networks with non-mixing channels that are more precise than predicted from coarse SF-TF biases or groups.

The GMM classes are widely distributed in all tested areas (***Figure 4—figure supplement 1B***). We constructed an inter V1-HVA connectivity matrix for the 65 classes (***Figure 4F***). The connection weight is determined by the proportion of pairs exhibiting high NCs. To investigate the modular structure of this network, we performed community detection analysis using the Louvain algorithm (***Rubinov and Sporns, 2010***). This analysis assigned densely connected nodes to the same module (***Figure 4F***). Overall, the connectivity matrix was split into four community modules (***Figure 4F***; ***Figure 4—figure supplement 1C***). Interestingly, the corresponding nodes in V1 and HVAs within each community

module have similar direction and SF-TF preferences (*Figure 4F*). For example, the module 2 nodes exhibited narrow vertical direction tuning and preferred high SF and low TF. Module 1 exhibited high SF preference without direction bias. Area differences in the characteristic tuning selectivity of each module are small, suggesting that the GMM class channels are common across the V1-HVA network. This is consistent with the overall broadcasting projection structure of V1 neurons (*Han et al., 2018*).

In summary, V1 and HVA neurons can be classified by their selectivity to oriented gratings, and they form precise, discrete channels or sub-networks. These sub-networks of neuron pairs with high NCs preserve selectivity by limiting inter-channel mixing. The organization of V1-HVA sub-networks exhibited properties consistent with those of V1-HVA feedforward projections, where the number of high-fidelity connections, rather than the strength of the connections, accounted for SF-TF biases among HVAs. Moreover, the precision of these networks extends beyond prior observations of general SF-TF biases to include orientation and direction tuning.

## Functional connectivity is stable across stimuli

Functional connectivity can be dynamic and transient, which complicates its relationship with structural (i.e. anatomical) connectivity, yet can provide more accurate predictions for network dynamics than the latter (*Randi et al., 2023*). We performed additional analysis to determine whether the NC-based functional connectivity analysis we performed above provides fundamental insights into neuron circuits beyond a stimulus-specific transient. We compared NC measurements in response to drifting gratings ($NC_{grat}$) to NC measurements in response to naturalistic videos ($NC_{nat}$). This analysis was restricted to the subset of neurons that responded to both types of stimuli in a separate set of experiments.

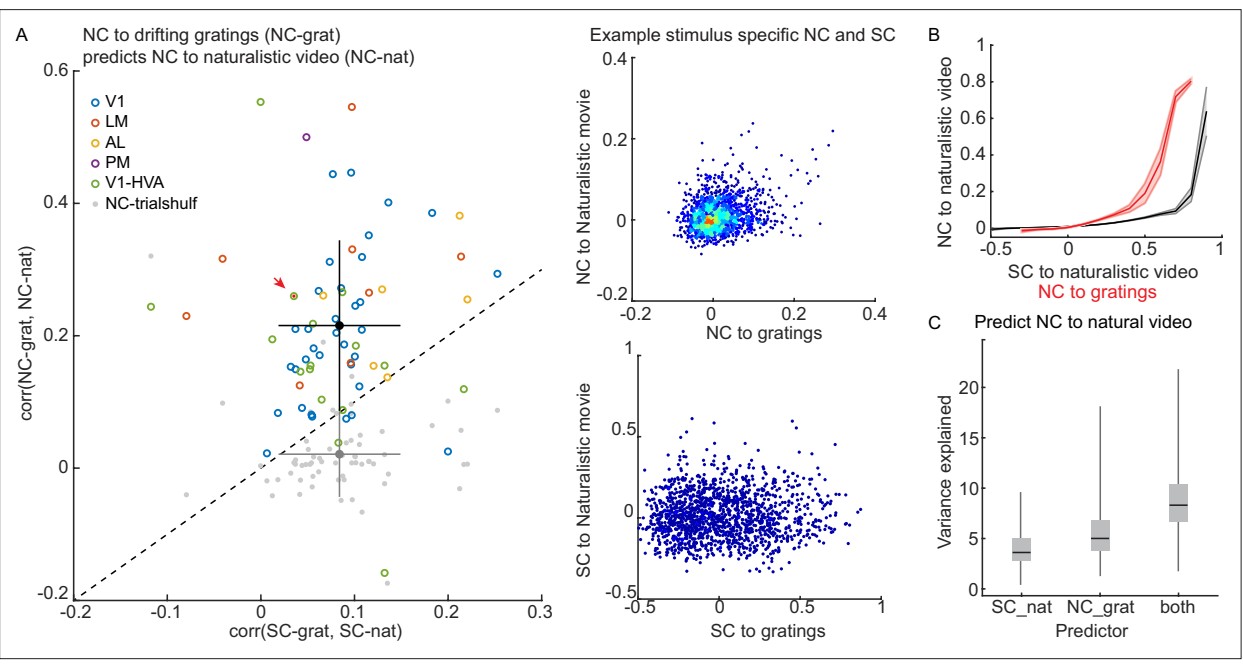

**Figure 5.** Noise correlations (NCs) across different classes of stimuli are more stable than tuning, or signal correlations (SCs). (**A**) (*left*) NCs measured during the naturalistic video are well correlated with NCs measured during drifting grating stimuli. The correlation between NCs across different stimuli is significantly higher than the correlation between corresponding SCs ($corr(NC_{grat}, NC_{nat}) = 0.22 \pm 0.13$; $corr(SC_{grat}, SC_{nat}) = 0.084 \pm 0.065$; t-test, p < 0.0001). Colored circles represent individual experiments. Gray dots represent trial-shuffled control ($corr(NCshulf_{grat}, NCshulf_{nat}) = 0.02 \pm 0.06$). The black/gray dot and error bars indicate the mean and SD for NC and SC. (*Right*) The correlation between (top) NCs and (bottom) SCs during grating and naturalistic video stimuli in an example dataset (red arrowhead in the lefthand plot). (**B**) NCs to a naturalistic video are positively related to the SCs, as well as to the NCs to drifting gratings. The shaded area indicates SEM. (**C**) The percentage of $NC_{nat}$ variance is explained by a linear model of $SC_{nat}$, $NC_{gray}$, or both factors. $NC_{gray}$ is a better linear predictor compared to $SC_{nat}$ ($NC_{gray}$, 5.3±3%, 4±2%; t-test, p<0.0001). Combining both factors predicts the $NC_{nat}$ even better (8±3%; t-test, p<0.0001). Variance explained is measured by R² of the linear regression.

The online version of this article includes the following figure supplement(s) for figure 5:

**Figure supplement 1.** Arousal modulation does not explain the NC connectivity.

So far, we have shown that SC (i.e. neuron tuning similarity) is the best predictor for NCs. However, a neuron pair that shares a high SC to drifting gratings does not guarantee a high SC to naturalistic videos ($corr(SC_{grat}, SC_{nat}) = 0.084 \pm 0.065$). Thus, it is reasonable to expect that NCs in response to gratings do not predict the NCs in response to naturalistic videos. However, we were surprised to find that the correlation between NCs to the two stimuli is significantly higher than that of the corresponding SCs ($corr(NC_{grat}, NC_{nat}) = 0.22 \pm 0.13$; **Figure 5A**). Thus, NCs across stimulus types are more predictable than SCs across stimulus types. To our knowledge, this is the first time this has been reported.

We used SCs to naturalistic videos ($SC_{nat}$), and gratings $NC_{grat}$ to predict $NC_{nat}$ using linear regressions. Both predictors are positively related to the $NC_{nat}$ (**Figure 5B**). We found that NC to gratings outperformed SC to naturalistic videos in predicting NC to naturalistic videos (t-test, p<0.0001; **Figure 5C**). Meanwhile, combining both predictors adds almost linearly in predicting NC to natural videos (**Figure 5C**), suggesting that the cross-stimulus NC predictor adds an independent dimension to the SC predictor. These results are evidence that NC-assessed functional connectivity reflects a fundamental aspect of the architecture of neuronal circuitry that is independent of visual input.

## Functional connectivity is not explained by the arousal state

Previous studies in awake mice suggested that a significant portion of the single-trial variance of visual cortical neural activity can be attributed to implicit behavioral or arousal factors during spontaneous locomotion activity (**Stringer et al., 2019**; **Dadarlat and Stryker, 2017**). We wondered whether the cross-stimulus stability of NCs observed in the current study would be explained by top-down modulation or the arousal state of the animal. The arousal state of the animal can be assessed using pupil dynamics (**Reimer et al., 2014**). Here, we characterized the arousal state by a multidimensional matrix composed of pupil centroid, pupil area, and principal components (PC) of pupil video (Materials and methods; **Figure 5—figure supplement 1A**). We measured the contribution of the multidimensional arousal factors to the variance of population neural activity.

We found that neuron activity is highly stimulus-driven while the animal is in a quiet state of wakefulness. The stimulus accounted for $30 \pm 11\%$ of the single trial variance of neural activity. Meanwhile, the multidimensional arousal factors explained about $3.5 \pm 2\%$, which is a significantly smaller contributor than the stimulus (p<0.0001, paired t-test; **Figure 5—figure supplement 1B**). The visual stimulus is more than ten times more influential over the trial-to-trial variability in activity than the arousal factors (**Figure 5—figure supplement 1C**).

Moreover, independent of the arousal factors and the visual stimulus, about $6 \pm 4\%$ of the activity of each neuron can be predicted by neuronal activity in its local population (greater than the contribution from arousal factors; p<0.0001, paired t-test; Materials and methods; **Figure 5—figure supplement 1B**). We reproduced the cross-stimulus comparison of SCs and NCs (**Figure 5A**) using just the residual neural activity after subtracting the arousal-modulated portion. The cross-stimulus stability of NC connectivity is preserved in the residual neural activity (**Figure 5—figure supplement 1D**). These results provided additional evidence that the cross-stimulus stable NC network relies on the circuitry of the neuron population rather than the bottom-up stimulus-driven activity or top-down modulation.

Notice that the contribution of arousal factors to population neural activity in the current dataset is smaller than the reported value from mice engaged in locomotion activity (**Stringer et al., 2019**). In one study (**Stringer et al., 2019**), a multidimensional arousal and facial dynamics matrix accounted for 21% of the variance of spontaneous neural activity. The difference could be due to the different brain states of the mice. In the current study, mice were in quiet wakefulness. The fluctuation of the arousal state during quiet wakefulness is much smaller than that during locomotion activity (**Reimer et al., 2014**). Moreover, spontaneous facial activity-correlated population neural activity reported in the previous studies (**Stringer et al., 2019**) does not indicate a causal relationship. In the next section, we tested the potential behavioral or brain state contribution to the modulation of functional networks in a causal simulation.

## Recurrent connection contributes to the stability of the NC network

After we observed the surprising cross-stimulus stability of the NC-based functional connectivity, we investigated potential underlying mechanisms. NCs can be due to both shared input (**Shadlen and Newsome, 1998**) and direct/indirect wiring (**Doiron et al., 2016**). Indeed, using a simple model with

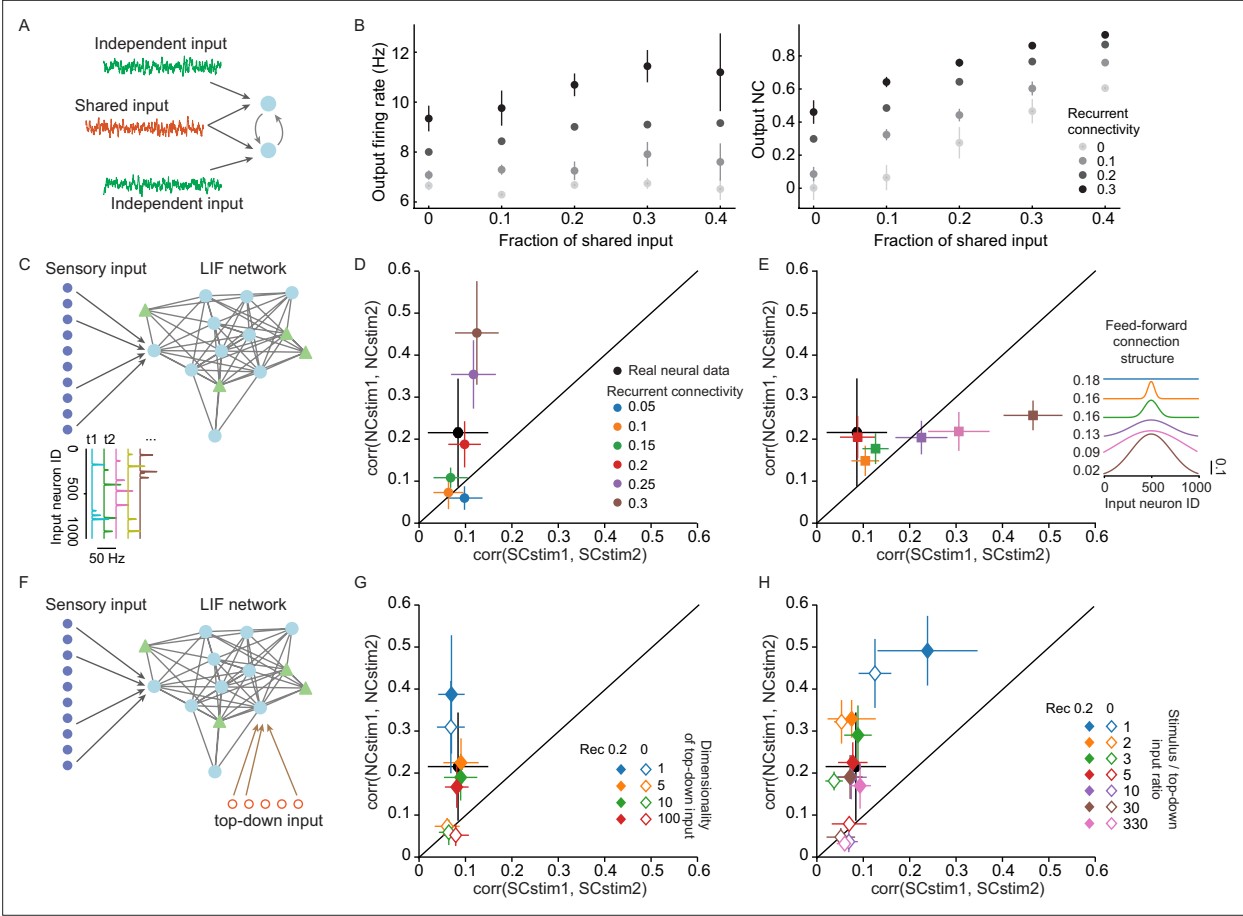

**Figure 6.** A network simulation shows that recurrent connectivity can contribute to the stability of the NC network. (**A**) A model with two leaky integrate and fire (LIF) neurons that are connected through excitatory synapses. The LIF neurons receive a fraction of shared input (red) and independent input (green) from a Poisson input layer. (**B**) The firing rate (left) and NC (right) of the two LIF neurons in a toy model (**A**) is regulated by the fraction of shared input and the strength of the recurrent connection. (**C**) Schematic of an LIF neuron network model with randomly connected LIF neurons and an input Poisson layer. The structure of the input connection and the strength of the recurrent connection are modulated in the simulation (**D, E**). (**D**) In networks with random input connection structures, increasing recurrent connection strength leads to higher cross-stimulus stability of the NC network. Among the values tested, recurrent connection = 0 .2 (red) generated a network that was closest to the mouse L2/3 visual neurons (black). (**E**) In simulations with 0.2 recurrent connectivity strength, regulating the input structure does not change the cross-stimulus stability of the NC network but leads to higher cross-stimulus stability of the SC network. (**F**) Same LIF network as C with an additional source of top-down input. The dimensionality and strength of the top-down input are modulated in the simulation (**G, H**). (**G**) In a network that is mainly driven by sensory input and receives moderate top-down modulation, the sensory/top-down input ratio is 10, recurrent connection is required to reproduce the NC network's cross-stimulus pattern. Changing the dimensionality of top-down input generated similar NC network patterns (overlapping dots in the figure). (**H**) In networks that receive strong top-down input, the sensory/top-down input ratio is no greater than 1.6, top-down modulation alone can generate cross-stimulus stable NC. Error bars in all panels indicate the SD of multiple randomly initialized simulations under the same condition.

two leaky integrate-and-fire (LIF) neurons, we found that the NC is positively regulated by a larger fraction of shared input as well as by the increasing recurrent connection strength (*Figure 6A and B*).

We then asked how the two sources contribute to the cross-stimulus stability of the NC functional network using LIF neuronal network simulations (*Figure 6C*). The simulated neuronal network contains 80 excitatory neurons and 20 inhibitory neurons that are randomly connected. The input layer contains 1000 independent Poisson spiking neurons. The network parameters are determined based on previous work (*Song et al., 2000*) and all the simulations generated comparable LIF firing rates (4–6 Hz), as well as NCs (population mean: 0.05–0.25) and SC values (population mean: 0.01–0.15).

In the first set of simulations, the feedforward (FFD) connection from the input layer to the LIF network is random. Increasing recurrent connection strengths (ranging from 0.05 to 0.3) generated NC-based networks with higher cross-stimulus stability (*Figure 6D*). A recurrent connection strength of 0.2 best reproduced the mouse data. In the second batch of simulations, we fixed the recurrent

connection strength to 0.2 but manipulated the input FFD connection structure ranging from random to increasingly wider bell shapes (*Figure 6E*). This means that the local neurons received increasingly similar FFD input. We found that increasingly similar local FFD input did not lead to higher NC stability, but did increase SC similarity across stimuli (*Figure 6E*). Also, the random FFD input connection structure (0.18 FFD, red) reproduced the experimentally observed NC network the best (*Figure 6E*). Thus, these LIF simulations showed that although both shared input and recurrent connections contributed to the NC, the recurrent connections are critical for generating the observed cross-stimulus stability of the NC functional network.

As top-down modulation was suggested to contribute to neural activity, we introduced additional inputs to the LIF network to explore the interplay between top-down inputs and NC-based network structure (*Figure 6F*). We compared the cross-stimulus stability of NC and SC in top-down modulated LIF networks with or without recurrent connections. When the LIF network receives moderate strength top-down inputs and strong stimulus input, recurrent connectivity is required to reproduce the cross-stimulus stable NC connectivity for multidimensional top-down modulation. Note that in the present data set, we found that the stimulus/top-down input ratio is about 10 (*Figure 5—figure supplement 1C*). This is true for cases when the top-down input is multidimensional (*Figure 6G*), and both the current study and a previous study found (*Stringer et al., 2019*) that indeed top-down input to the mouse visual cortex is high dimensional (*Figure 5—figure supplement 1B*). On the other hand, when top-down input is unrealistically strong (the stimulus/top-down input ratio is no greater than three, instead of the ten-fold factor we find in our data), the LIF network can generate cross-stimulus stable NC connectivity with or without recurrent connections (*Figure 6H*). However, as stated above, in the current study, the visual cortical neuron population is highly stimulus-driven (10-fold greater than top-down input; *Figure 5—figure supplement 1B*). Together, these LIF simulations suggested that recurrent connections are critical for generating the observed cross-stimulus stable NC connectivity.

## Discussion

We used large-scale two-photon calcium imaging across cortical areas to show that neuron-resolution, NC-based assessments of functional connectivity exhibited tuning-specific organization, across millimeter length scales. We provided a detailed analysis of the rigor of measuring NCs with calcium imaging and assessed their reliability given the imprecision of calcium imaging for inferring spiking activity. The connectivity we observed reinforces data from axonal projection patterns (*Glickfeld et al., 2013*; *Han et al., 2018*; *Kim et al., 2018*) and provides more granular functional resolution, down to >60 tuning classes we found with GMM analysis. Thus, V1 broadcasts high-fidelity channels of information to HVAs. The projections preserve fidelity by minimizing the mixing among tuning channels. Such a connectivity pattern readily supports the generation of segregated tuning biases in mouse HVAs (*Yu et al., 2022*; *Marshel et al., 2011*). However, it remains unclear whether this supports the generation of new tuning features in HVAs that do not exist in V1, which would require feature integration (*Juavinett and Callaway, 2015*). Thus, further studies in anatomical and functional connectivity are needed to address the issue of feature integration.

NCs are an activity-based trace of connectivity in neuronal networks. NCs can provide specific insights about neuronal circuit architecture when analyzed rigorously (*Cohen and Kohn, 2011*; *Schulz et al., 2015*; *Ecker et al., 2014*; *Doiron et al., 2016*). For example, researchers have proposed quantitative hypotheses for establishing a link between NCs and information encoding (*Kohn et al., 2016*; *Moreno-Bote et al., 2014*; *Kanitscheider et al., 2015*; *Rumyantsev et al., 2020*; *Hazon et al., 2022*; *Kafashan et al., 2021*), and information transmission (*Zandvakili and Kohn, 2015*; *Ohiorhenuan et al., 2010*), as well as neuron tuning (*Ecker et al., 2011*; *Panzeri et al., 2022*). NCs offer a data-driven complement to model-driven approaches for analyzing neural circuitry (*Pillow et al., 2008*; *Keeley et al., 2020*; *Goris et al., 2014*; *Rabinowitz et al., 2015*), and approaches that require cellular resolution manipulations (*Randi et al., 2023*; *Oldenburg et al., 2024*). Technological advancements continue to enable further access to anatomical network structures (*Turner et al., 2022*; *Gao et al., 2022*; *Velicky et al., 2023*), but since it is difficult to predict the emergent behavior of a collection of neurons from purely anatomical information, functional connectivity, or NC in particular, provides a valuable bridge.

To aid in generating a mechanistic interpretation of NC, one strategy is to compare NC across different states (*Doiron et al., 2016*), such as anesthetized versus awake (*Ecker et al., 2014*), walking

versus stationary (*Dadarlat and Stryker, 2017*), spontaneous versus stimulus-driven conditions (*Miller et al., 2014*), and different task conditions (*Ruff and Cohen, 2016*). In the current study, we compared the NC of the mouse visual cortex to two different visual stimuli. Our findings of cross-stimulus stability of NC provide the first evidence showing that NC outperforms SC in predicting the functional neural network to a different visual stimulus in the mouse visual cortex. It is encouraging that despite limitations in NC measurement and uncertainty regarding its source, the NC-SC relationship is stable and provides effective constraints to neuron network models with stimulus input, recurrent connections, and multidimensional top-down modulation. NC characterizes the activation structure of a functional neuron network. Ensemble analysis serves a similar purpose by directly identifying a subset of coactivated neurons. Previous ensemble analyses have suggested that the co-firing patterns of neurons in response to a visual stimulus are highly similar to those observed during spontaneous co-firing events (*Pérez-Ortega et al., 2021*; *Miller et al., 2014*). Both the ensemble analysis and the NC analysis emphasized the critical role of intrinsic network interactions, rather than bottom-up or top-down inputs, in generating the emergent behavior of neuron populations.

Recent breakthroughs in systems neuroscience have been made possible by advancements in large-scale population neuron recording techniques (*Yu et al., 2021*; *Stirman et al., 2016*; *Papadopouli et al., 2024*; *Manley et al., 2024*; *Siegle et al., 2021*). Interpreting the circuit mechanisms underlying the collective behavior of large neuron populations is challenging but offers significant opportunities for understanding the brain (*Urai et al., 2022*). In this study, we demonstrated that characterizing the functional connectivity, specifically the noise correlation (NC), of the neuron population provides valuable insights into interpreting the underlying circuit mechanisms. However, a quantitative model linking functional connectivity with the emergent activity of the neuron population is still missing. This motivates future studies in computational modeling and experimentally probing the emergent activity of neuron populations using behavior or artificial methods.

## Materials and methods

### Animals and surgery

All animal procedures and experiments were approved by the Institutional Animal Care and Use Committee of the University of North Carolina at Chapel Hill or the University of California Santa Barbara and carried out in accordance with the regulations of the US Department of Health and Human Services. GCaMP6s expressing transgenic adult mice of both sexes were used in this study. Mice were 110–300 days old for data collection. GCaMP6s expressing were induced by the triple crossing of TITL-GCaMP6s line (Allen Institute Ai94), Emx1-Cre line (Jackson Labs #005628), and ROSA:LNL:tTA line (Jackson Labs #011008; *Madisen et al., 2015*). Mice were housed under a 12 hr/12 hr light-dark cycle, and experiments were performed during the dark cycle of mice. Mice were anesthetized with isoflurane (1.5–1.8%) and acepromazine (1.5–1.8 mg/kg body weight) when performing visual cortex craniotomy. Carprofen (5 mg/kg body weight) was administered prior to surgery. The body temperature was maintained using physically activated heat packs during surgery. Mouse eyes were kept moist with ointment during surgery. The scalp overlaying the right visual cortex was removed, and a custom head-fixing imaging chamber with a 5 mm diameter opening was mounted to the skull with cyanoacrylate-based glue (Oasis Medical) and dental acrylic (Lang Dental). A 4 mm diameter craniotomy was performed over the visual cortex and covered with #1 thickness coverslip.

### Locating visual areas with intrinsic signal optical imaging (ISOI)

ISOI experiments were carried out similarly as previously (*Stirman et al., 2016*; *Smith et al., 2017*; *Smith and Trachtenberg, 2007*). Briefly, the pial vasculature images and intrinsic signal images were collected using a CCD camera (Teledyne DALSA 1M30) at the craniotomy window. A 4.7×4.7 mm² cortical area was imaged at $9.2\mu m$/pixel spatial resolution and at 30 Hz frame rate. The pial vasculature was illuminated and captured through green filters (550±50 nm and 560±5 nm, Edmund Optics). The ISO image was collected by focusing $600\mu m$ down from the pial surface. The intrinsic signals were illuminated and captured through red filters (700±38 nm, Chroma and 700±5 nm, Edmund Optics). Custom ISOI instruments were adapted from *Kalatsky and Stryker, 2003*. Custom acquisition software for ISOI imaging collection was adapted from David Ferster (*Stirman et al., 2016*). During ISOI, mice were 20 cm from a flat monitor (60×34 cm²), which covers the visual field (110° x 75°) of the left

eye. Mice were lightly anesthetized with isoflurane (0.5%) and acepromazine (1.5–3 mg/kg). The body temperature was maintained at 37 °C using a custom electric heat pad (*Stirman et al., 2016*). Intrinsic signal response to the vertical and horizontal drifting bar was used to generate azimuth and elevation retinotopic maps (*Figure 1—figure supplement 1A*). The retinotopic maps were then used to locate V1 and HVAs. Borders between these areas were drawn at the meridian of elevation and azimuth retinotopy manually (*Marshel et al., 2011*; *Smith et al., 2017*). The vasculature map then provided landmarks to identify visual areas in two-photon imaging.

## In vivo two-photon calcium imaging

Two-photon imaging was carried out using a custom Trepan2p microscope controlled by custom LabView software (*Stirman et al., 2016*). Simultaneous dual-region imaging was achieved by splitting the excitation beam and temporally multiplexing laser pulses (*Stirman et al., 2016*). Two-photon excitation light from an 80 MHz Ti:Sapph laser (Newport Spectra-Physics Mai Tai DeepSee) was split into two beams through polarization optics, and one path was delayed 6.25 ns relative to the other. The two beams were independently directed with custom voice-coil actuated steering mirrors and tunable lenses, such that the X, Y, Z planes of the two paths are independently positioned within the full field (4.4 mm diameter). Both beams were scanned by the resonant scanner (4 kHz, Cambridge Technologies), and a single photon signal was collected by a photomultiplier tube (PMT; H7422P-40, Hamamatsu), and demultiplexed using outboard electronics prior to digitization. In the current study, two-photon imaging regions of 500 x 500 $\mu m^2$ were collected at 13.3 Hz for two-region imaging or 6.67 Hz for quad-region imaging. Imaging was performed with <80 mW of excitation (910 nm) laser power, as measured out of the front of the objective. Mice recovered in their home cage for at least 2 days after surgery, before acquiring two-photon imaging. Mice were head-fixed ~11 cm from a flat monitor, with their left eye facing the monitor, during imaging. Approximately 70° x 45° of the left visual field was covered. If not otherwise stated, two-photon images were recorded from quiet awake mice. For anesthetized experiments, mice were lightly anesthetized under 1% isoflurane.

In a subset of experiments, we monitored mouse pupil position and diameter using a custom-controlled CCD camera (GigE, National Instruments) at 20–25 fps. No additional light stimulation was used for pupil imaging. Pupil dynamics was analyzed using a custom-written imaging processing code in Matlab.

## Visual stimuli

Visual stimulation was displayed on a 60 Hz LCD monitor (9.2x15 cm$^2$). All stimuli were displayed in full contrast. For coarse population RF and single neuron RF mapping (*Figure 1—figure supplement 1B–D*), a rectangular (7.5° x 8.8°) bright moving patch containing vertical drifting grating (2 Hz, 0.05 cpd) on a dark background was displayed. The moving patch appeared and disappeared on a random position of the full monitor in pseudo-random order without interruption by a gray screen and was presented on each position for 5 s.

To characterize the value and structure of the correlation of V1 and HVAs, we showed mice full-screen sine-wave drifting grating stimuli in eight directions (0–315°, in 45° step), with an SF of 0.02, 0.05, or 0.19 cpd, and a TF of 1, 2, or 8 Hz (72 conditions in total). Each of the sine-wave drifting grating stimuli was presented for 2 s in pseudo-random order. Stimuli with the same SF and TF were presented successively without interruption. A gray screen was presented for 3 s when changing the SF and TF of stimuli. Each stimulus was presented 7–25 times, 15.2 trials on average.

In a subset of experiments, we also characterized the cross-stimulus stability of functional networks using combo stimuli with naturalistic videos and full contrast drifting gratings (at 2 Hz, 0.05 cpd). Two naturalistic videos, each lasting for 32 s were generated by navigating a mouse home cage using a GoPro camera. Each stimulus was presented for 10–20 times, 19.8 trials on average, interleaved with an 8 s gray screen.

## Calcium imaging processing

Calcium imaging processing was carried out using custom MATLAB code (*Yu et al., 2022*). Two-photon calcium imaging was motion corrected using Suite2p subpixel registration module (*Pachitariu et al., 2016*). Neuron ROIs and cellular calcium traces were extracted from imaging stacks using custom code adapted from Suite2p modules (*Pachitariu et al., 2016*). Neuropil contamination was corrected

by subtracting the common time series (1st PC) of a spherical surrounding mask of each neuron from the cellular calcium traces (*Harris et al., 2016*). Neuropil contamination-corrected calcium traces were then deconvolved using a Markov chain Monte Carlo (MCMC) method (*Pnevmatikakis et al., 2013*). For each calcium trace, we repeated the MCMC simulation for 400 times and measured the signal-to-noise of MCMC spike train inference for each cell. For all subsequent analyses, only cells that had reliable spike train inference results were included. Neurons with low responsiveness were excluded for subsequent analysis (trial-averaged spike count to preferred spatiotemporal frequency summed over all orientations < 1; or trial-averaged spike count to a 32 s naturalistic video <1).

## Receptive field analysis

We mapped RFs by reverse correlation of neuronal responses with the locations of the moving patch of drifting grating stimulus. For population RF mapping, population neuronal responses of simultaneously recorded neurons from a 500 x 500 µm$^2$ imaging window were reverse correlated with the stimulus locations. We found the best fit elliptical 2D Gaussian profile for the intensity map ($F$) of the single neuron RF or the population neuron RF by minimizing the least square error.

$$F = A * exp(-M(\frac{(x - x_0)^2}{2\sigma_x^2} + \frac{(y - y_0)^2}{2\sigma_y^2}))$$

The amplitude ($A$), rotation matrix ($M$), centroid ($x_0, y_0$), and spread ($\sigma_x, \sigma_y$) of the 2D Gaussian are found through the least square fit. The long and short axes of the 2D Gaussian of single-neuron RF are estimated by $\sqrt{2ln2} * \sigma$, which are the half-width half-maximal of the Gaussian spread. Similarly, population RF is characterized by the full-width half-maximal of the best-fit 2D Gaussian spread (average over the long and short axes), and the size of the 2D Gaussian profile above the half-maximal.

## Gaussian mixture model

To characterize the tuning properties without an investigator bias, we used a data-driven approach and neurons were clustered using a Gaussian mixture model (GMM) based on the trial-averaged responses to the drifting gratings. Only reliable responsive neurons were included for GMM analysis (trial-to-trial Pearson correlation of the inferred spike trains >0.08, spike trains were binned at 500 ms). Neuronal responses of the whole population pooled over all tested areas were first denoised and reduced dimension by minimizing the prediction error of the trial-averaged response using the PCs. 45 PCs were kept for population responses to the drifting gratings. We also tested a wide range number of PCs (20-70), and we found the tuning group clustering was not affected by the number of PCs. Neurons collected from different visual areas and different animals were pooled together in training GMM. GMMs were trained using MATLAB build function *fitgmdist* with a wide range number of clusters. A model of 65 classes was selected based on the Bayesian information criterion. We also examined models with smaller (20, 30, and 45) or even larger numbers of classes (75), the overall results held regardless of the number of GMM classes. *Figure 1—figure supplement 2A* shows the t-SNE embedding of GMM classes and the direction selectivity of each class. The size of each GMM class is shown in *Figure 1—figure supplement 1F*. t-SNE was performed using MATLAB built-in function *tsne*, with Mahalanobis distance method.

To summarize the spatial-temporal tuning properties of neurons (*Figure 1E*), we manually organized the 65 GMM classes into 6 groups based on their preferred SF-TF (*Figure 1—figure supplement 2A*). Group 1 prefers low SF and low TF (LSLT, 0.02 cpd, 1–2 Hz), group 2 prefers medium SF and low TF (MSLT, 0.05 cpd, 1–2 Hz), group 3 prefers high SF low TF (HSLT, 0.19 cpd, 1–2 Hz), group 4 prefers low SF high TF (LSHT, 0.02 cpd, 8 Hz), group 5 prefers medium SF and high TF (MSHT, 0.05 cpd, 8 Hz), group 6 prefers high SF and high TF (HSHT, 0.19 cpd, 8 Hz), and group 7 is not specific (*Figure 1—figure supplement 2B*). Group 7 included four classes that did not exhibit specific response features, among them two classes are extremely small (each contains <5 neurons), and the other two contain neurons with small response strength (mean spike count <0.5 spikes/s to preferred stimulus). As we have been inclusive in data selection for the GMM training and included low-firing neurons, the latter two classes contain about 1500 neurons. It is justifiable to exclude low-response neurons from further analysis. Thus, the whole group 7 was excluded for further analysis.

## GMM classification accuracy

We examined the accuracy of GMM classification for neuron responses to drifting gratings. We performed GMM clustering on 10 random subsets of neurons (90% of all neurons) and measured the clustering consistency. First, we measured the consistency of the centers of the Gaussian clusters, which are 45D vectors in the PC dimensions. We measured the Pearson correlation of Gaussian center vectors independently defined by GMM clustering on random subsets of neurons. We found the center of the Gaussian profile of each class was consistent (*Figure 1C*). The same class of different GMMs was identified by matching the center of the class. Then, we asked whether a neuron was classified in the same class in each GMM model. We found neurons were consistently classified into the same class in GMMs of a random subset of data (*Figure 1D*). We also performed GMM on population data after randomly shuffling neuron identity (10 permutations). Classes were identified by matching the center of the class and then grouped following the previous definition. We found that neurons are allocated into the same SF-TF group in GMMs of randomly ordered data (*Figure 1—figure supplement 1G*, H). These analyses suggested that GMM provided a reliable classification of neurons.

## Orientation and direction selectivity

The direction and orientation selectivity of each neuron was computed using the spatiotemporal frequency of drifting grating stimuli that drove the strongest response for that neuron. The direction selectivity index and orientation selectivity index were computed using the following equations.

$$SI = \frac{(R_{prefer} - R_{null})}{(R_{prefer} + R_{null})} \tag{1}$$

The polar plots of tuning groups were generated by averaging responses to the preferred direction of each neuron within a tuning group and normalized to one (*Figure 1—figure supplement 3C*). For neurons with high direction selectivity, neuron responses to the preferred direction were considered, while for neurons with low direction selectivity (DSI <0.5), neuron responses to both preferred and null directions were included.

## ISOI warping

We spatially registered the ISOI map of V1 to align with that of LM or AL (*Figure 3—figure supplement 1C–H*). We first segmented the ISOI map by color segmentation using K-means clustering and then determined the center of each color segment. Then we performed the affine transformation of color band centers of V1 to match that of LM or AL. The transformation matrix M was determined by minimizing the distance between transformed V1 centers and LM or AL centers using Matlab function *fminsearch*.

## Correlation calculation

Signal correlation was defined as the correlation of stimulus tuning between neurons. The stimulus tuning of each neuron was estimated by the mean response of trials to the same stimulus ($\overline{Y}_i$), $i$ is the neuron identity.

$$r_{sig} = \frac{cov(\overline{Y}_i, \overline{Y}_j)}{\sqrt{var(\overline{Y}_i) * var(\overline{Y}_j)}}$$

Noise correlation was defined as the trial-to-trial correlation of residual spike count (1 s time window, if not otherwise stated) after subtracting the mean response to each stimulus of the 72-condition sine-wave drifting gratings. Residual spike count to all stimuli (e.g. gratings with different directions and SFs and TFs), and all trials were concatenated into one column vector per neuron ($u_i$, $u_i = Y_i - \overline{Y}_i$). The noise correlation $r_{sc}$ was computed as the Pearson correlation of $u_i$ and $u_j$.

$$r_{sc} = \frac{cov(u_i, u_j)}{\sqrt{var(u_i) * var(u_j)}}$$

$i, j$ indicates neuron identity. Signal correlation was defined as the neuron-to-neuron Pearson correlation of mean responses. Mean response was a 72-element column vector, computed by trial

averaging responses to sine-wave gratings with 72 conditions. To examine the relation between noise correlation and joint firing rate between a pair of neurons, we computed the mean joint spike count (geometric mean spike count average over all stimuli).

We computed inter-area NCs with simultaneously recorded regions that shared greater than 40% of population RF. We kept this criterion even though we did not detect a relationship between the inter-area NC and the fraction of population RF overlap within the tested range (p=0.37).

## Fidelity of noise correlation measurement

### Tolerance of correlation calculation to inaccuracy in spike train inference

We quantify the spike train inference accuracy using a previously published data set with simultaneous cell-attached recording and two-photon imaging of GCamp6s from mouse V1 (*Chen et al., 2013*; http://crcns.org/data-sets/methods/cai-1). We performed spike train inference on the recordings with stable baseline and good correspondence between calcium trace and electrophysiology recording (linear correlation, *r*>0.1; bin 0.1 s; *Figure 2—figure supplement 1A, B*). The signal-to-noise (SNR) of the calcium trace of the calibration data is 12.3±5. It is comparable with the SNR of the calcium signal of the current study (8.7±1.8).

We further evaluated how the correlation calculation was affected by inaccurate spikes train recovery. We took publicly available electrophysiology recordings of mouse V1 neurons (*Theis et al., 2016*; http://spikefinder.codeneuro.org/), and computed residual spike count correlation at 1 s time bin after perturbations on the ground truth spikes train. We did four types of perturbations: (1) randomly missing spikes; (2) missing isolated spikes as the signal-to-noise of the calcium signal of isolated spikes may be low; (3) missing all spikes within a burst; (4) missing 60% spikes within a burst (*Figure 2—figure supplement 1C*). We identified isolated spikes or burst spikes by thresholding the inter-spike interval of each spike. A spike that was >*t* s away from spikes flanking itself was a *t* isolated spike. A spike that was <*t* s away from another spike was a *t* burst spike. The residual spike count correlation computed with perturbed spike trains was linearly correlated with ground truth (*Figure 2—figure supplement 1D*) and exhibited good tolerance to up to 60% missing spikes by all types of perturbation (fidelity > 0.6; *Figure 2—figure supplement 1E*).

### Significance of noise correlation

Since the value of noise correlations was small, we tested whether these values were significantly above zero. We compared the noise correlation with trial-shuffled noise correlation; the latter was computed using trial-shuffled data (the order of trials was randomized for each neuron independently). The population-mean noise correlation computed with trial-aligned data was significantly higher than that of the trial-shuffled data with the size of the experimental population (*Figure 2B*).

### Accuracy of noise correlation

We investigated the accuracy of noise correlation estimation with both data and model. The individual noise correlations of the same set of neurons varied when computing using a different random subset of trials (*Figure 2D*). We computed the population mean value of the noise correlation of a random subset of neuron pairs and calculated the confidence interval for estimating the population mean noise correlation. The accuracy of population-mean estimation increases with the number of neurons, even with a limited number of trials (*Figure 2C*). We further characterized the estimation accuracy by simulating correlated neuron population (*Macke et al., 2009*), which allows an arbitrary number of trials. The expected firing rate and expected population mean correlation match our experimental data. To achieve an accurate estimation (1/10 standard error/mean value) of the population mean correlation converges with >100 neurons even using experimental level trial numbers (*Figure 2C*).

## Community module analysis

We constructed a V1-HVA connectivity matrix using the fraction of high NC (NC >mean + 2.5*SD of trial-shuffled NC) pairs between each GMM class. We performed community detection analysis using the Louvain algorithm (*Rubinov and Sporns, 2010*), which assigned densely connected nodes to the same module. The spatial smooth parameter $\gamma$ that generated the largest deviation from a random connectivity matrix is picked. The analysis was performed using the Brain Connectivity Toolbox (brain-connectivity-toolbox.net).

## Leaky integrate-and-fire neuron network simulation

LIF simulations were carried out using the Brian2 simulation engine in Python (*Stimberg et al., 2019*). The LIF neuron network model was defined similarly as *Song et al., 2000*. In brief, the membrane potential of LIF neurons was given by the equation below:

$$\frac{dv}{dt} = (ge * (Ee - v) + gi * (Ei - v) + El - v)/\tau_m$$

where $\tau_m$ corresponds to the membrane time constant (20ms). $ge$, $gi$, and $Ee$, $Ei$ are the excitatory and inhibitory synaptic conductance and their respective reversal potential ($Ee$=0 mV, $Ei$ = –80 mV). The membrane potential was simulated with a time resolution of $dt = 0.1ms$. $El$ (–70 mV) corresponds to the resting potential. The dynamics of synaptic conductance were given by exponential decay functions $ge/dt = -ge/\tau_e$ and $gi/dt = -gi/\tau_i$ where $\tau_e$ (5ms) and $\tau_i$ (10ms) are the decay time constants for excitatory and inhibitory synapses.

Synaptic connections between LIF neurons occurred with probability $p = 0.02$, and the strength of the connections is defined as $W_{ij}^{rec}$, ($i,j$ indicate source and target LIF neuron ID, $i \cong j$). $W_{ij}^{rec} = J_{max}$ or 0 if not connected. In simulations for *Figure 6D* $J_{max}$ ranges from 0.05 to 0.3, while in simulations for *Figure 6E–H* $J_{max}$ is fixed to 0.2 for closed markers or 0 for open markers.

The LIF network received feedforward (FFD) stimulus input from Poisson neurons (N=1,000 in the network simulation, and N=80 in the toy model), whose firing follows time-varying Poisson processes (0-30 Hz; *Figure 6C* insert). The instant firing rate of the FFD Poisson neuron is defined by combining five weighted Gaussian profiles. The firing rate of the FFD input neuron varies for different types of stimuli by randomly initializing the weights to the five Gaussian profiles. In simulations with top-down input, we adjusted the FFD firing rate by changing the amplitude of the Gaussian profile for moderate or strong top-down input while keeping the LIF network firing rate to 4-6 Hz.

The Poisson FFD input neurons are connected to LIF neurons with a probability of $p = 0.2$, and the strength of the connections is defined as $W_{lj}^{ff} \cdot W_{lj}^{ff} = S_{lj}$ or 0 if not connected. $l$ indicates the source FFD input neuron.

$$S_{l,j} = a * 0.2 * exp(-((i/10.0 - j)/b)^2) + c * 0.2$$

a, b, and c are parameters that manipulate the structure of the FFD connection, ranging from fully random to fully bell-shaped (*Figure 6E*). FFD connection set to random in simulations for *Figure 6D, G and H*.

Top-down inputs in *Figure 6G–H* are generated by $d$ independent Poisson process. $d$ defines the dimensionality of the top-down input, ranging from 1 to 100 for *Figure 6G* and fixed $d = 10$ for *Figure 6H*. The instant firing rate of these neurons is pooled from a uniform random distribution and changes on each trial. The firing rate of these neurons ranges from 1 to 150 Hz for generating moderate to strong top-down input to the LIF neurons. The top-down input neurons are randomly connected with the LIF network with a probability of $p = 0.2$. The strength of the top-down connections is defined as $W_{kj}^{td} \cdot W_{kj}^{td} = G_{kj}$ or 0 if not connected. $G_{kj}$ is generated by a uniform distribution between 0 and 0.2 that indicates the source top-down input neuron. The stimulus/top-down input ratio is determined by the changes in membrane potential induced by each source of input.

In the toy model, the connectivity $S_{lj}$ equals the fraction of shared input. The Poisson input neuron fires constantly at 5 Hz.

## Decompose neural activity

We use linear fitting to estimate the fraction of population neural activity explained by stimulus input, pupil dynamics, and network interactions (*Figure 5—figure supplement 1B*). First, we estimated the variance of population neural activity due to stimulus input. The expected stimulus-response of each neuron is estimated by the average response of trials to the same stimulus. The contribution of the stimulus-driven portion of population neural activity is defined by the equation below, or the variance of the expected stimulus response divided by the total variance ($i$ is stimulus ID; $t$ is trial ID).

$$Frac_{stim} = Var(\overline{Y_i})/Var(Y_{it})$$

After subtracting the stimulus-driven portion of population neural activity ($Y_{res} = Y_{it} - \overline{Y_i}$), we next measure the contribution from pupil dynamics ($X_{pupil}$) to population neural activity using a reduced rank regression model. The relationship between residual population neural activity and pupil dynamics is defined as

$$Y_{res} = X_{pupil} * B$$

The reduced rank objective can be written as

$$\min_{B} tr[(Y_{res} - X_{pupil}B)(Y_{res} - X_{pupil}B)^T]$$

$$s.t.\, rank(B) <= r$$

The coefficient of the reduced rank regression model $B_c$ is estimated by

$$B_c = B_{OLS} * VV^T$$

$B_{OLS}$ is the ordinary least squares solution, and $\hat{Y}_{OLS} = X_{pupil} * B_{OLS}$. $V$ is the first $r$ eigenvectors of $\hat{Y}_{OLS}^T \hat{Y}_{OLS}$.

In this model, the pupil dynamics matrix is composed of pupil area, centroid position, and PCs of the pupil video that explain more than 1% of the variance of the video (**Figure 5—figure supplement 1A**), together they composed the factor matrix of $B$. On average, 11.2 PCs were kept and accounted for 87% of the variance of the video. Both the reduced rank regression model and the ordinary regression model generated quantitatively similar results. However, a reduced rank regression model is recommended to constrain the fitting better and avoid rank deficiency problems. The rank dimension is 13.4±3. With the solution of reduced rank regression ($B_c$) and prediction ($\hat{Y}_{RRR} = X_{pupil} * B_c$), we estimated the contribution of pupil dynamics to neural activity.

$$Frac_{pupil} = Var(\hat{Y}_{RRR})/Var(Y_{it})$$

Next, we measured the contribution of network interactions to the unexplained portion of neural activities, which was computed by subtracting the pupil-related firing from the residual neural activity ($Y_{unexp} = Y_{res} - \hat{Y}_{RRR}$). We model the unexplained portion of individual neural activity ($Y_{unexp}^{(n)}$ for target neuron $n$) with the rest of the network using a PC regression model. PCs of unexplained neural activity (exclude one target neuron at a time, $PC_{Score}^{(-n)}$) with > 0.1 variance were kept.

$$Y_{unexp}^{(n)} = PC_{Score}^{(-n)} * B_{pc}$$

The coefficients of the equation were estimated using an ordinary least squares solution. We estimated the neural activity that was generated by network interaction as $\hat{Y}_{net} = PC_{Score}^{(-n)} * B_{pc}$. The contribution of network interaction to neural activity is measured by

$$Frac_{net} = Var(\hat{Y}_{net})/Var(Y_{it})$$

## Acknowledgements

Funding was provided by grants from the NIH (R01EY024294, R01NS091335), the NSF (1707287, 1450824), the Simons Foundation (SCGB325407), and the McKnight Foundation to SLS; a Helen Lyng White Fellowship to YY; a career award from Burroughs Wellcome to JNS; and training grant support for CRD (T32NS007431).

## Additional information

### Competing interests
Spencer LaVere Smith: serves as a consultant for optics and neuroscience companies and is a founder of Pacific Optics. The other authors declare that no competing interests exist.

## Funding

| Funder | Grant reference number | Author |
| --- | --- | --- |
| NIH | R01EY024294 | Spencer LaVere Smith |
| NIH | R01NS091335 | Spencer LaVere Smith |
| NSF | 1707287 | Spencer LaVere Smith |
| NSF | 1450824 | Spencer LaVere Smith |
| Simons Foundation | SCGB325407 | Spencer LaVere Smith |
| McKnight Foundation | | Spencer LaVere Smith |
| Helen Lyng White Fellowship | | Yiyi Yu |
| Burroughs Wellcome Fund | | Jeffery N Stirman |
| NIH | T32NS007431 | Christopher R Dorsett |

The funders had no role in study design, data collection and interpretation, or the decision to submit the work for publication.

### Author contributions

Yiyi Yu, Conceptualization, Data curation, Software, Formal analysis, Validation, Investigation, Visualization, Methodology, Writing - original draft, Writing – review and editing; Jeffery N Stirman, Methodology; Christopher R Dorsett, Formal analysis; Spencer LaVere Smith, Conceptualization, Resources, Supervision, Funding acquisition, Methodology, Project administration, Writing – review and editing

### Author ORCIDs

Yiyi Yu https://orcid.org/0000-0003-1053-413X
Spencer LaVere Smith https://orcid.org/0000-0002-2021-7034

### Ethics

This study was performed in strict accordance with the recommendations in the Guide for the Care and Use of Laboratory Animals of the National Institutes of Health. All animal procedures and experiments were handled according to approved Institutional Animal Care and Use Committee (IACUC) protocols of the University of North Carolina at Chapel Hill or approved IACUC protocols (#950) of the University of California Santa Barbara and carried out in accordance with the regulations of the US Department of Health and Human Services. All surgery was performed under isoflurane anesthesia, and every effort was made to minimize suffering.

Reviewer #1 (Public review): https://doi.org/10.7554/eLife.97848.3.sa1
Reviewer #2 (Public review): https://doi.org/10.7554/eLife.97848.3.sa2
Reviewer #3 (Public review): https://doi.org/10.7554/eLife.97848.3.sa3
Author response https://doi.org/10.7554/eLife.97848.3.sa4

# Additional files

### Supplementary files

MDAR checklist

Supplementary file 1. A complete accounting of neurons used in the study, by recording areas and animal IDs.

### Data availability

Data and code generated in the current study are available on github https://github.com/yuyiyi/Broadcast_channel (copy archived at *Yu, 2026*).

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
