## [Editor Report · eLife assessment]

.This **important** study uses state-of-the-art, multi-region two-photon calcium imaging to characterize the statistics of functional connectivity between visual cortical neurons. Although alternative interpretations may partially account for the data, the study provides **solid** evidence that functionally distinct classes of neurons convey visual information via parallel channels within and across both primary and higher-order cortical areas.

---

## [Referee Report · Reviewer #1 (Public review)]

Summary:

Using multi-region two-photon calcium imaging, the manuscript meticulously explores the structure of noise correlations (NCs) across mouse visual cortex and uses this information to make inferences about the organization of communication channels between primary visual cortex (V1) and higher visual areas (HVAs). Using visual responses to grating stimuli, the manuscript identifies 6 tuning groups of visual cortex neurons, and finds that NCs are highest among neurons belonging to the same tuning group whether or not they are found in the same cortical area. The NCs depend on the similarity of tuning of the neurons (their signal correlations) but are preserved across different stimulus sets - noise correlations recorded using drifting gratings are highly correlated with those measured using naturalistic videos. Based on these findings, the manuscript concludes that populations of neurons with high NCs constitute discrete communication channels that convey visual signals within and across cortical areas.

Strengths:

Experiments and analyses are conducted to a high standard and the robustness of noise correlation measurements is carefully validated. To control for potential influences of behaviour-related top-down modulation of noise correlations, the manuscript uses measurements of pupil dynamics as a proxy for behavioural state and shows that this top-down modulation cannot explain the stability of noise correlations across stimuli.

Weaknesses:

The interpretation of noise correlation measurements as a proxy from network connectivity is fraught with challenges. While the data clearly indicate the existence of distributed functional ensembles, the notion of communication channels implies the existence of direct anatomical connections between them, which noise correlations cannot measure.

The traditional view of noise correlations is that they reflect direct connectivity or shared inputs between neurons. While it is valid in a broad sense, noise correlations may reflect shared top-down input as well as local or feedforward connectivity. This is particularly important since mouse cortical neurons are strongly modulated by spontaneous behavior (e.g. Stringer et al, Science, 2019). Therefore, noise correlation between a pair of neurons may reflect whether they are similarly modulated by behavioral state and overt spontaneous behaviors. Consequently, noise correlation alone cannot determine whether neurons belong to discrete communication channels.

---

## [Referee Report · Reviewer #2 (Public review)]

Summary:

This groundbreaking study characterizes the structure of activity correlations over millimeter scale in the mouse cortex with the goal of identifying visual channels, specialized conduits of visual information that show preferential connectivity. Examining the statistical structure of visual activity of L2/3 neurons, the study finds pairs of neurons located near each other or across distances of hundreds of micrometers with significantly correlated activity in response to visual stimuli. These highly correlated pairs have closely related visual tuning sharing orientation and/or spatial and/or temporal preference as would be expected from dedicated visual channels with specific connectivity.

Strengths:

The study presents best-in-class mesoscopic-scale 2-photon recordings from neuronal populations in pairs of visual areas (V1-LM, V1-PM, V1-AL, V1-LI). The study employs diverse visual stimuli that capture some of the specialization and heterogeneity of neuronal tuning in mouse visual areas. The rigorous data quantification takes into consideration functional cell groups as well as other variables that influence trial-to-trial correlations (similarity of tuning, neuronal distance, receptive field overlap, behavioral state). The paper demonstrates the robustness of the activity clustering analysis and of the activity correlation measurements. The paper shows convincingly that the correlation structure observed with grating stimuli is present in the responses to naturalistic stimuli. A simple simulation is provided that suggest that recurrent connectivity is required for the stimulus invariance of the results. The paper is well written and conceptually clear. The figures are beautiful and clear. The arguments are well laid out and the claims appear in large part supported by the data and analysis results (but see weaknesses).

Weaknesses:

An inherent limitation of the approach is that it cannot reveal which anatomical connectivity patterns are responsible for observed network structure. A methodological issue that does not seem completely addressed is whether the calcium imaging measurements with their limited sensitivity amplify the apparent dependence of noise correlations on the similarity of tuning. Although the paper shows that noise correlation measurements are robust to changes in firing rates / missing spikes, the effects of receptive field tuning dissimilarity are not addressed directly. The calcium responses of mouse visual cortical neurons are sharply tuned. Neurons with dissimilar receptive fields may show too little overlap in their estimated firing rates to infer noise correlations, which could lead to underestimation of correlations across groups of dissimilar neurons.

---

## [Referee Report · Reviewer #3 (Public review)]

Summary:

Yu et al harness the capabilities of mesoscopic 2P imaging to record simultaneously from populations of neurons in several visual cortical areas and measure their correlated variability. They first divide neurons in 65 classes depending on their tuning to moving gratings. They found the pairs of neurons of the same tuning class show higher noise correlations (NCs) both within and across cortical areas. Based on these observations and a model they conclude that visual information is broadcast across areas through multiple, discrete channels with little mixing across them.

NCs can reflect indirect or direct connectivity, or shared afferents between pairs of neurons, potentially providing insight on network organization. While NCs have been comprehensively studied in neurons pairs of the same area, the structure of these correlations across areas is much less known. Thus, the manuscripts present novel insights on the correlation structure of visual responses across multiple areas.

Strengths:

The measurements of shared variability across multiple areas are novel. The results are mostly well presented and many thorough controls for some metrics are included.

Weaknesses:

I have concerns that the observed large intra class/group NCs might not reflect connectivity but shared behaviorally driven multiplicative gain modulations of sensory evoked responses. In this case, the NC structure might not be due to the presence of discrete, multiple channels broadcasting visual information as concluded. I also find that the claim of multiple discrete broadcasting channels needs more support before discarding the alternative hypothesis that a continuum of tuning similarity explains the large NCs observed in groups of neurons.

Specifically:

Major concerns:

(1) Multiplicative gain modulation underlying correlated noise between similarly tuned neurons

(1a) The conclusion that visual information is broadcasted in discrete channels across visual areas relies on interpreting NC as reflecting, direct or indirect connectivity between pairs, or common inputs. However, a large fraction of the activity in the mouse visual system is known to reflect spontaneous and instructed movements, including locomotion and face movements, among others. Running activity and face movements are one of the largest contributors to visual cortex activity and exert a multiplicative gain on sensory evoked responses (Niell et al , Stringer et al, among others). Thus, trial-by-fluctuations of behavioral state would result in gain modulations that, due to their multiplicative nature, would result in more shared variability in cotuned neurons, as multiplication affects neurons that are responding to the stimulus over those that are not responding (see Lin et al , Neuron 2015 for a similar point).

In the new version of the manuscript, behavioral modulations are explicitly considered in Figure S8. New analyses show that most of the variance of the neuronal responses is driven by the stimulus, rather than by behavioural variable. However, they new analyses still do not address if the shared noise correlation in cotuned neurons is also independent of behavioral modulations .

As behavioral modulations are not considered this confound affects the conclusions and the conclusion that activity in communicated unmixed across areas (results in Figure 4), as it would result in larger NCs the more similar the tuning of the neurons is, independently of any connectivity feature. It seems that this alternative hypothesis can explain the results without the need of discrete broadcasting channels or any particular network architecture and should be addressed to support the main claims.

(2) Discrete vs continuous communication channels

(2a) One of the author's main claims is that the mouse cortical network consists of discrete communication channels, as stated in teh title of the paper. This discreteness is based on an unbiased clustering approach on the tuning of neurons, followed by a manual grouping into six categories with relation to the stimulus space. I believe there are several problems with this claim. First, this clustering approach is inherently trying to group neurons and discretise neural populations. To make the claim that there are 'discrete communication channels' the null hypothesis should be a continuous model. An explicit test in favor of a discrete model is lacking, i.e. are the results better explained using discrete groups vs. when considering only tuning similarity? Second, the fact that 65 classes are recovered (out of 72 conditions) and that manual clustering is necessary to arrive at the six categories is far from convincing that we need to think about categorically different subsets of neurons. That we should think of discrete communication channels is especially surprising in this context as the relevant stimulus parameter axes seem inherently continuous: spatial and temporal frequency. It is hard to motivate the biological need for a discretely organized cortical network to process these continuous input spaces.

Finally, as stated in point 1, the larger NCs observed within groups than across groups might be due to the multiplicative gain of state modulations, due to the larger tuning similarity of the neurons within a class or group.

---

## [Author Response]

The following is the authors’ response to the original reviews.

General Response

We are grateful for the constructive comments from reviewers and the editor.

The main point converged on a potential alternative interpretation that top-down modulation to the visual cortex may be contributing to the NC connectivity we observed. For this revision, we address that point with new analysis in Fig. S8 and Fig. 6. These results indicate that top-down modulation does not account for the observed NC connectivity.

We performed the following analyses.

(1) In a subset of experiments, we recorded pupil dynamics while the mice were engaged in a passive visual stimulation experiment (Fig. S8A). We found that pupil dynamics, which indicate the arousal state of the animal, explained only 3% of the variance of neural dynamics. This is significantly smaller than the contribution of sensory stimuli and the activity of the surrounding neuronal population (Fig. S8B). In particular, the visual stimulus itself typically accounted for 10-fold more variance than pupil dynamics (Fig. S8C). This suggests that the population neural activity is highly stimulus-driven and that a large portion of functional connectivity is independent of top-down modulation. In addition, after subtracting the neural activity from the pupil-modulated portion, the cross-stimulus stability of the NC was preserved (Fig. S8D).

We note that the contribution from pupil dynamics to neural activity in this study is smaller than what was observed in an earlier study (Stringer et al. 2019 Science). That can be because mice were in quiet wakefulness in the current study, while mice were in spontaneous locomotion in the earlier study. We discuss this discrepancy in the main text, in the subsection *“Functional connectivity is not explained by the arousal state”*.

(2) We performed network simulations with top-down input (Fig. 6F-H). With multidimensional top-down input comparable to the experimental data, recurrent connections within the network are necessary to generate cross-stimulus stable NC connectivity (Fig. 6G). It took increasing the contribution from the top-down input (i.e., to more than 1/3 of the contribution from the stimulus), before the cross-stimulus NC connectivity can be generated by the top-down modulation (Fig. 6H). Thus, this analysis provides further evidence that top-down modulation was not playing a major role in the NC connectivity we observed.

These new results support our original conclusion that network connectivity is the principal mechanism underlying the stability of functional networks.

**Public Reviews:**
**Reviewer #1 (Public Review)**:Using multi-region two-photon calcium imaging, the manuscript meticulously explores the structure of noise correlations (NCs) across the mouse visual cortex and uses this information to make inferences about the organization of communication channels between primary visual cortex (V1) and higher visual areas (HVAs). Using visual responses to grating stimuli, the manuscript identifies 6 tuning groups of visual cortex neurons and finds that NCs are highest among neurons belonging to the same tuning group whether or not they are found in the same cortical area. The NCs depend on the similarity of tuning of the neurons (their signal correlations) but are preserved across different stimulus sets - noise correlations recorded using drifting gratings are highly correlated with those measured using naturalistic videos. Based on these findings, the manuscript concludes that populations of neurons with high NCs constitute discrete communication channels that convey visual signals within and across cortical areas.Experiments and analyses are conducted to a high standard and the robustness of noise correlation measurements is carefully validated. However, the interpretation of noise correlation measurements as a proxy from network connectivity is fraught with challenges. While the data clearly indicates the existence of distributed functional ensembles, the notion of communication channels implies the existence of direct anatomical connections between them, which noise correlations cannot measure.The traditional view of noise correlations is that they reflect direct connectivity or shared inputs between neurons. While it is valid in a broad sense, noise correlations may reflect shared top-down input as well as local or feedforward connectivity. This is particularly important since mouse cortical neurons are strongly modulated by spontaneous behavior (e.g. Stringer et al, Science, 2019). Therefore, noise correlation between a pair of neurons may reflect whether they are similarly modulated by behavioral state and overt spontaneous behaviors. Consequently, noise correlation alone cannot determine whether neurons belong to discrete communication channels.Behavioral modulation can influence the gain of sensory-evoked responses (Niell and Stryker, Neuron, 2010). This can explain why signal correlation is one of the best predictors of noise correlations as reported in the manuscript. A pair of neurons that are similarly gain-modulated by spontaneous behavior (e.g. both active during whisking or locomotion) will have higher noise correlations if they respond to similar stimuli. Top-down modulation by the behavioral state is also consistent with the stability of noise correlations across stimuli. Therefore, it is important to determine to what extent noise correlations can be explained by shared behavioral modulation.

We thank the reviewer for the constructive and positive feedback on our study.

The reviewer acknowledged the quality of our experiments and analysis and stated a concern that the noise correlation can be explained by top-down modulation. We have addressed this concern carefully in the revision, please see the General Response above.

**Reviewer #2 (Public Review):**
Summary:This groundbreaking study characterizes the structure of activity correlations over a millimeter scale in the mouse cortex with the goal of identifying visual channels, specialized conduits of visual information that show preferential connectivity. Examining the statistical structure of the visual activity of L2/3 neurons, the study finds pairs of neurons located near each other or across distances of hundreds of micrometers with significantly correlated activity in response to visual stimulation. These highly correlated pairs have closely related visual tuning sharing orientation and/or spatial and/or temporal preference as would be expected from dedicated visual channels with specific connectivity.Strengths:The study presents best-in-class mesoscopic-scale 2-photon recordings from neuronal populations in pairs of visual areas (V1-LM, V1-PM, V1-AL, V1-LI). The study employs diverse visual stimuli that capture some of the specialization and heterogeneity of neuronal tuning in mouse visual areas. The rigorous data quantification takes into consideration functional cell groups as well as other variables that influence trial-to-trial correlations (similarity of tuning, neuronal distance, receptive field overlap). The paper convincingly demonstrates the robustness of the clustering analysis and of the activity correlation measurements. The calcium imaging results convincingly show that noise correlations are correlated across visual stimuli and are strongest within cell classes which could reflect distributed visual channels. A simple simulation is provided that suggests that recurrent connectivity is required for the stimulus invariance of the results. The paper is well-written and conceptually clear. The figures are beautiful and clear. The arguments are well laid out and the claims appear in large part supported by the data and analysis results (but see weaknesses).Weaknesses:An inherent limitation of the approach is that it cannot reveal which anatomical connectivity patterns are responsible for observed network structure. The modeling results presented, however, suggest interestingly that a simple feedforward architecture may not account for fundamental characteristics of the data. A limitation of the study is the lack of a behavioral task. The paper shows nicely that the correlation structure generalizes across visual stimuli. However, the correlation structure could differ widely when animals are actively responding to visual stimuli. I do think that, because of the complexity involved, a characterization of correlations during a visual task is beyond the scope of the current study.An important question that does not seem addressed (but it is addressed indirectly, I could be mistaken) is the extent to which it is possible to obtain reliable measurements of noise correlation from cell pairs that have widely distinct tuning. L2/3 activity in the visual cortex is quite sparse. The cell groups laid out in Figure S2 have very sharp tuning. Cells whose tuning does not overlap may not yield significant trial-to-trial correlations because they do not show significant responses to the same set of stimuli, if at all any time. Could this bias the noise correlation measurements or explain some of the dependence of the observed noise correlations on signal correlations/similarity of tuning? Could the variable overlap in the responses to visual responses explain the dependence of correlations on cell classes and groups?With electrophysiology, this issue is less of a problem because many if not most neurons will show some activity in response to suboptimal stimuli. For the present study which uses calcium imaging together with deconvolution, some of the activity may not be visible to the experimenters. The correlation measure is shown to be robust to changes in firing rates due to missing spikes. However, the degree of overlap of responses between cell pairs and their consequences for measures of noise correlations are not explored.Beyond that comment, the remaining issues are relatively minor issues related to manuscript text, figures, and statistical analyses. There are typos left in the manuscript. Some of the methodological details and results of statistical testing also seem to be missing. Some of the visuals and analyses chosen to examine the data (e.g., box plots) may not be the most effective in highlighting differences across groups. If addressed, this would make a very strong paper.

We thank the reviewer for acknowledging the contributions of our study.

We agree with the reviewer that future studies on behaviorally engaged animals are necessary. Although we also agree with the reviewer that behavior studies are out the scope of the current manuscript, we have included additional analysis and discussion on whether and how top-down input would affect the NC connectivity in the revision. Please see the General Response above.

**Reviewer #3 (Public Review):**
Summary:Yu et al harness the capabilities of mesoscopic 2P imaging to record simultaneously from populations of neurons in several visual cortical areas and measure their correlated variability. They first divide neurons into 65 classes depending on their tuning to moving gratings. They found the pairs of neurons of the same tuning class show higher noise correlations (NCs) both within and across cortical areas. Based on these observations and a model they conclude that visual information is broadcast across areas through multiple, discrete channels with little mixing across them.NCs can reflect indirect or direct connectivity, or shared afferents between pairs of neurons, potentially providing insight on network organization. While NCs have been comprehensively studied in neuron pairs of the same area, the structure of these correlations across areas is much less known. Thus, the manuscripts present novel insights into the correlation structure of visual responses across multiple areas.Strengths:The study uses state-of-the art mesoscopic two-photon imaging.The measurements of shared variability across multiple areas are novel.The results are mostly well presented and many thorough controls for some metrics are included.Weaknesses:I have concerns that the observed large intra-class/group NCs might not reflect connectivity but shared behaviorally driven multiplicative gain modulations of sensory-evoked responses. In this case, the NC structure might not be due to the presence of discrete, multiple channels broadcasting visual information as concluded. I also find that the claim of multiple discrete broadcasting channels needs more support before discarding the alternative hypothesis that a continuum of tuning similarity explains the large NCs observed in groups of neurons.Specifically:Major concerns:(1) Multiplicative gain modulation underlying correlated noise between similarly tuned neurons(1a) The conclusion that visual information is broadcasted in discrete channels across visual areas relies on interpreting NC as reflecting, direct or indirect connectivity between pairs, or common inputs. However, a large fraction of the activity in the mouse visual system is known to reflect spontaneous and instructed movements, including locomotion and face movements, among others. Running activity and face movements are some of the largest contributors to visual cortex activity and exert a multiplicative gain on sensory-evoked responses (Niell et al, Stringer et al, among others). Thus, trial-by-fluctuations of behavioral state would result in gain modulations that, due to their multiplicative nature, would result in more shared variability in cotuned neurons, as multiplication affects neurons that are responding to the stimulus over those that are not responding (see Lin et al, Neuron 2015 for a similar point).As behavioral modulations are not considered, this confound affects most of the conclusions of the manuscript, as it would result in larger NCs the more similar the tuning of the neurons is, independently of any connectivity feature. It seems that this alternative hypothesis can explain most of the results without the need for discrete broadcasting channels or any particular network architecture and should be addressed to support its main claims.(1b) In Figure 5 the observations are interpreted as evidence for NCs reflecting features of the network architecture, as NCs measured using gratings predicted NC to naturalistic videos. However, it seems from Figure 5 A that signal correlations (SCs) from gratings had non-zero correlations with SCs during naturalistic videos (is this the case?). Thus, neurons that are cotuned to gratings might also tend to be coactivated during the presentation of videos. In this case, they are also expected to be susceptible to shared behaviorally driven fluctuations, independently of any circuit architecture as explained before. This alternative interpretation should be addressed before concluding that these measurements reflect connectivity features.

We thank the reviewer for acknowledging the contributions of our study.

The reviewer suggested that gain modulation might be interfering with the interpretation of the NC connectivity. We have addressed this issue in the General Response above.

Here, we will elaborate on one additional analysis we performed, in case it might be of interest. We carried out multiplicative gain modeling by implementing an established method (Goris et al. 2014 Nat Neurosci) on our dataset. We were able to perform the modeling work successfully. However, we found that it is not a suitable model for explaining the current dataset because the multiplicative gain induced a negative correlation. This seemed odd but can be explained. First, top-down input is not purely multiplicative but rather both additive and multiplicative. Second, the top-down modulation is high dimensional. Third, the firing rate of layer 2/3 mouse visual cortex neurons is lower than the firing rates for non-human primate recordings used in the development of the method (Goris et al. 2014 Nat Neurosci). Thus, we did not pursue the model further. We just mention it here in case the outcome might be of interest to fellow researchers.

(2) Discrete vs continuous communication channels(2a) One of the author's main claims is that the mouse cortical network consists of discrete communication channels. This discreteness is based on an unbiased clustering approach to the tuning of neurons, followed by a manual grouping into six categories in relation to the stimulus space. I believe there are several problems with this claim. First, this clustering approach is inherently trying to group neurons and discretise neural populations. To make the claim that there are 'discrete communication channels' the null hypothesis should be a continuous model. An explicit test in favor of a discrete model is lacking, i.e. are the results better explained using discrete groups vs. when considering only tuning similarity? Second, the fact that 65 classes are recovered (out of 72 conditions) and that manual clustering is necessary to arrive at the six categories is far from convincing that we need to think about categorically different subsets of neurons. That we should think of discrete communication channels is especially surprising in this context as the relevant stimulus parameter axes seem inherently continuous: spatial and temporal frequency. It is hard to motivate the biological need for a discretely organized cortical network to process these continuous input spaces.(2b) Consequently, I feel the support for discrete vs continuous selective communication is rather inconclusive. It seems that following the author's claims, it would be important to establish if neurons belong to the same groups, rather than tuning similarity is a defining feature for showing large NCs.

Thanks for pointing this out so that we can clarify.

We did not mean to argue that the tuning of neurons is discrete. Our conclusions are not dependent on asserting a particular degree of discreteness. We performed GMM clustering to label neurons with an identity so that we could analyze the NC connectivity structure with a degree of granularity supported by the data. Our analysis suggested that communication happens within a class, rather than through mixed classes. We realized that using the term “discrete” may be confusing. In the revised text we used the term “unmixed” or “non-mixing” instead to emphasize that the communication happens between neurons belonging to the same tuning cluster, or class.

However, we do see how the question of discreteness among classes might be interesting to readers. To provide further information, we have included a new Fig. S2 to visualize the GMM classes using t-SNE embedding.

Finally, as stated in point 1, the larger NCs observed within groups than across groups might be due to the multiplicative gain of state modulations, due to the larger tuning similarity of the neurons within a class or group.

We have addressed this issue in the General Response above and the response to comment (1).

**Recommendations for the authors:**

**Reviewing Editor (Recommendations For The Authors):**
A general recommendation discussed with the reviewers is to make use of behavioural recording to assess whether shared behaviourally driven modulations can explain the observed relation between SC and NC, independently of the network architecture. Alternatively, a simulation or model might also address this point as well as the possibility that the relation of SC and NC might be also independent of network architecture given the sparseness of the sensory responses in L2/3.

We have addressed this in the General Response above.

Broadly speaking, inferring network architecture based on NCs is extremely challenging. Consequently, the study could also be substantially improved by reframing the results in terms of distributed co-active ensembles without insinuation of direct anatomical connectivity between them.

We agree that the inferring network architecture based on NCs is challenging. The current study has revealed some principles of functional networks measured by NCs, and we showed that cross-stimulus NC connectivity provides effective constraints to network modeling. We are explicit about the nature of NCs in the manuscript. For example, in the Abstract, we write “to measure correlated variability (i.e., noise correlations, NCs)”, and in the Introduction, we write “NCs are due to connectivity (direct or indirect connectivity between the neurons, and/or shared input)”. We are following conventions in the field (e.g., Sporns 2016; Cohen and Kohn 2011).

Notice also that the abstract or title should make clear that the study was made in mice.

Sorry for the confusion, we now clearly state the study was carried out in mice in the Abstract and Introduction.

**Reviewer #1 (Recommendations For The Authors):**
The manuscript presents a meticulous characterization of noise correlations in the visual cortical network. However, as I outline in the public review, I think the use of noise correlations to infer communication channels is problematic and I urge the authors to carefully consider this terminology. Language such as "strength of connections" (Figure 4D) should be avoided.

We now state in the figure legend that the plot in Fig. 4D shows the average NC value.

My general suggestion to the authors, which primarily concerns the interpretation of analyses in Figures 4-6, is to consider the possible impact of shared top-down modulation on noise correlations. If behavioral data was recorded simultaneously (e.g. using cameras to record face and body movements), behavioral modulation should be considered alongside signal correlation as a possible factor influencing NCs.

We have addressed this issue in the General Response above.

I may be misunderstanding the analysis in Figure 4C but it appears circular. If the fraction of neurons belonging to a particular tuning group is larger, then the number of in-group high NC pairs will be higher for that group even if high NC pairs are distributed randomly. Can you please clarify? I frankly do not understand the analysis in Figure 4D and it is unclear to me how the analyses in Figure 4C-D address the hypotheses depicted in the cartoons.

Sorry for the confusion, we have clarified this in the Fig. 4 legend.

Each HVA has a SFTF bias (Fig. 1E,F; Marshel et al., 2011; Andermann et al., 2011; Vries et al., 2020). Each red marker on the graph in Fig. 4C is a single V1-HVA pair (blue markers are within an area) for a particular SFTF group (Fig. 1). The x-axis indicates the number of high NC pairs in the SFTF group in the V1-HVA pair divided by the total number of high NC pairs per that V1-HVA pair (summed over all SFTF groups). The trend is that for HVAs with a bias towards a particular SFTF group, there are also more high NC pairs in that SFTF group, and thus it is consistent with the model on the right side. This is not circular because it is possible to have a SFTF bias in an HVA and have uniformly low NCs. The reviewer is correct that a random distribution of high NCs could give a similar effect, which is still consistent with the model: that the number of high NC pairs (and not their specific magnitudes) can account for SFTF biases in HVAs.

To contrast with that model, we tested whether the average NC value for each tuning group varies. That is, can a small number of very high NCs account for SFTF biases in HVAs? That is what is examined in Fig. 4D. We found that the average NC value does not account for the SFTF biases. Thus, the SFTF biases were not related to the modulation in NC (i.e., functional connection strength).

I found the discussion section quite odd and did not understand the relevance of the discussion of the coefficient of variation of various quantities to the present manuscript. It would be more useful to discuss the limitations and possible interpretations of noise correlation measurements in more detail.

We have revised the discussion section to focus on interpreting the results of the current study and comparing them with those of previous studies.

Figure 3B: please indicate what the different colors mean - I assume it is the same as Figure 3A but it is unclear.

We added text to the legend for clarification.

Typos: Page 7: "direct/indirection wiring", Page 11: "pooled over all texted areas"

We have fixed the typos.

**Reviewer #2 (Recommendations For The Authors):**
The significance of the results feels like it could be articulated better. The main conclusion is that V1 to HVA connections avoid mixing channels and send distinctly tuned information along distinct channels - a more explicit description of what this functional network understanding adds would be useful to the reader.

Thanks for the suggestion. We have edited the introduction section and the discussion section to make the take-home message more clear.

Previous studies with anatomical data already indicate distinctly tuned channels - several of which the authors cite - although inconsistently:• Kim et al 2018 https://doi.org/10.1016/j.neuron.2018.10.023• Glickfeld et al., 2013 (cited)• Han et al., 2022 (cited)• Han and Bonin 2023 (cited)

Thanks for the suggestion, we now cite the Kim et al. 2018 paper.

I think the information you provide is valuable - but the value should be more clearly spelled out - This section from the end of the discussion for example feels like abdicates that responsibility:"In summary, mesoscale two-photon imaging techniques open up the window of cellular-resolution functional connectivity at the system level. How to make use of the knowledge of functional connectivity remains unclear, given that functional connectivity provides important constraints on population neuron behavior."A discussion of how the results relate to previous studies and a section on the limitations of the study seems warranted.

Thanks for the suggestion, we have extensively edited the discussion section to make the take-home message clear and discuss prior studies and limitations of the present study.

Details:Analyses or simulations showing that the dependency of correlations on similarity of tuning is not an artifact of how the data was acquired is in my mind missing and if that is the case it is crucial that this be addressed.

At each step of data analysis, we performed control analysis to assess the fidelity of the conclusion. For example, on the spike train inference (Fig. S4), GMM clustering (Fig. S1), and noise correlation analysis (Figs. 2, S5).

None of the statistical testing seems to use animals as experimental units (instead of neurons). This could over-inflate the significance of the results. Wherever applicable and possible, I would recommend using hierarchical bootstrap for testing or showing that the differences observed are reproducible across animals.

We analyzed the tuning selectivity of HVAs (Fig. 1F) using experimental units, rather than neurons. It is very difficult to observe all tuning classes in each experiment, so pooling neurons across animals is necessary for much of the analysis. We do take care to avoid overstating statistical results, and we show the data points in most figure to give the reader an impression of the distributions.

Page 2. "The number of neurons belonged to the six tuning groups combined: V1, 5373; LM, 1316; AL, 656; PM, 491; LI, 334." Yet the total recorded number of neurons is 17,990. How neurons were excluded is mentioned in Methods but it should be stated more explicitly in Results.

We have added text in the Fig. 1 legend to direct the audience to the Methods section for information on the exclusion / inclusion criteria.

Figure 1C, left. I don't understand how correlation is the best way to quantify the consistency of class center with a subset of data. Why not use for example as the mean square error. The logic underlying this analysis is not explained in Methods.

Sorry for the confusion, we have clarified this in the Methods section.

We measured the consistency of the centers of the Gaussian clusters, which are 45-dimensional vectors in the PC dimensions. We measured the Pearson correlation of Gaussian center vectors independently defined by GMM clustering on random subsets of neurons. We found the center of the Gaussian profile of each class was consistent (Fig. 1C). The same class of different GMMs was identified by matching the center of the class.

Figure 1E. There are statements in the text about cell groups being more represented in certain visual areas. These differences are not well represented in the box plots. Can't the individual data points be plotted? I have also not found the description and results of statistical testing for these data.

We have replotted the figure (now Fig. 1F) with dot scatters which show all of the individual experiments.

Figure 2A, right, since these are paired data, I am not quite sure why only marginal distributions are shown. It would be interesting to know the distributions of correlations that are significant.

This is only for illustration showing that NCs are measurable and significantly different from zero or shuffled controls. The distribution of NCs is broad and has both positive and negative values. We are not using this for downstream analysis.

Figure 4A, I wonder if it would not be better to concentrate on significant correlations.

We focused on large correlation values rather than significant values because we wanted to examine the structure of “strongly connected” neuron pairs. Negative and small correlation values can be significant as well. Focusing on large values would allow us to generate a clear interpretation.

Figure 4B, 'Mean strength of connections' which I presume mean correlations is not defined anywhere that I can see.

I believe the reviewer means Fig. 4D. It means the average NC value. We have edited the figure legend to add clarity.

Figure 4F, a few words explaining how to understand the correlation matrix in text or captions would be helpful.

Sorry for the confusion, we have clarified this part in figure legend for Fig. 4F.

Page 5, right column: Incomplete sentence: "To determine whether it is the number of high NC pairs or the magnitude of the NCs,".

We have edited this sentence.

Page 5, right column: "Prior findings from studies of axonal projections from V1 to HVAs indicated that the number of SF-TF-specific boutons -rather than the strength of boutons- contribute to the SF-TF biases among HVAs (Glickfeld et al., 2013)." Glickfeld et al. also reported that boutons with tuning matched to the target area showed stronger peak dF/F responses.

Thank you. We have revised this part accordingly.

Page 9, the Discussion and Figure 7 which situates the study results in a broader context is welcome and interesting, but I have the feeling that more words should be spent explaining the figure and conceptual framework to a non-expert audience. I am a bit at a loss about how to read the information in the figure.

Sorry for the confusion, we have added an explanation about this section (page 10, right column).

As far as I can see, data availability is not addressed in the manuscript. The data, code to analyze the data and generate the figures, and simulation code should be made available in a permanent public repository. This includes data for visual area mapping, calcium imaging data, and any data accessory to the experiments.

We have stated in the manuscript that code and data are available upon request. We regularly share data with no conditions (e.g., no entitlement to authorship), and we often do so even prior to publication.

The sex of the mice should be indicated in Figure T1.

The sex of the mice was mixed. This is stated in the Methods section.

Methods:Section on statistical testing, computation of explained variance missing, etc. I feel many analyses are not thoroughly described.

Sorry for the confusion, we have improved our method section.

Signal correlation (similarity between two neurons' average responses to stimuli) and its relation to noise correlation is not formally defined.

We have included the definition of signal correlation in the Methods.

Number of visual stimulation trials is not stated in Methods. Only stated figure caption.

The number of visual stimulus trials is provided in the last paragraph of the Methods section (Visual Stimuli).

Fix typos: incorrect spelling, punctuation, and missing symbols (e.g. closing parentheses).

We have carefully examined the spelling, punctuation, and grammar. We have corrected errors and we hope that none remain.

Why use intrinsic imaging to locate retinotopic boundaries in mice already expressing GCaMP6s?

We agree with the reviewer that calcium imaging of visual cortex can be used to identify the visual cortex.

It is true that areas can be mapped using the GCaMP signals. That is not our preferred approach. Using intrinsic imaging to define the boundary between V1 and HVAs has been a well refined routine in our lab for over a decade. It is part of our standard protocol. One advantage is that the data (from intrinsic signals) is of the same nature every time. This enables us to use the same mapping procedure no matter what reporters mice might be expressing (and the pattern, e.g., patchy or restricted to certain cell types).

**Reviewer #3 (Recommendations For The Authors):**
The possibilty that larger intra-group NCs observed simply reflect a multiplicative gain on cotuned neurons could be addressed using pupil and/or face recordings: Does pupil size or facial motion predict NCs and if factored out, does signal correlation still predict NCs?Perhaps a variant of the network model presented in Figure 6 with multiplicative gain could also be tested to investigate these issues.

We have addressed this issue in general response.

Here, we will elaborate on one additional analysis we performed, in case it might be of interest. We carried out multiplicative gain modeling by implementing an established method (Goris et al. 2014 Nat Neurosci) on our dataset. We were able to perform the modeling work successfully. However, we found that it is not a suitable model for explaining the current dataset because the multiplicative gain induced a negative correlation. This seemed odd but can be explained. First, top-down input is not purely multiplicative but rather both additive and multiplicative. Second, the top-down modulation is high dimensional. Third, the firing rate of layer 2/3 mouse visual cortex neurons is lower than the firing rates for non-human primate recordings used in the development of the method (Goris et al. 2014 Nat Neurosci). Thus, we did not pursue the model further. We just mention it here in case the outcome might be of interest to fellow researchers.

Similarly further analyses can be done to strengthen support for the claims that the observed NCs reflect discrete communication channels. A direct test of continuous vs categorical channels would strengthen the conclusions. One possible analysis would be to compare pairs with similar tuning (same SC) belonging to the same or different groups.

Thanks for pointing this out so that we can clarify.

We did not mean to argue that the tuning of neurons is discrete. Our conclusions are not dependent on asserting a particular degree of discreteness. We performed GMM clustering to label neurons with an identity so that we could analyze the NC connectivity structure with a degree of granularity supported by the data. Our analysis suggested that communication happens within a class, rather than through mixed classes. We realized that using the term “discrete” may be confusing. In the revised text we used the term “unmixed” or “non-mixing” instead to emphasize that the communication happens between neurons belonging to the same tuning cluster, or class.

However, we do see how the question of discreteness among classes might be interesting to readers. To provide further information, we have included a new Fig. S2 to visualize the GMM classes using t-SNE embedding.

I also found many places where the manuscript needs clarification and /or more methodological details:• How many times was each of the stimulus conditions repeated? And how many times for the two naturalistic videos? What was the total duration of the experiments?

The number of visual stimulus trials is provided in the last paragraph of the Methods section entitled *Visual Stimuli.* About 15 trials were recorded for each drifting grating stimulus, and about 20 trials were recorded for each naturalistic video.

• Typo: Suit2p should be Suite2p (section Calcium image processing - Methods).

We have fixed the typo.

• What do the error bars in Figure 1E represent? Differences in group representation across areas from Figure 1E are mentioned in the text without any statistical testing.

We have revised the Figure 1E (current Fig. 1F), and we now show all data points.

• The manuscript would benefit from a comparison of the observed area-specific tuning biases across areas (Figure 1E and others) with the previous literature.

We have included additional discussion on this in the last paragraph of the section entitled *Visual cortical neurons form six tuning groups*.

• Why are inferred spike trains used to calculate NCs? Why can't dF/F be used? Do the results differ when using dF/F to calculate NC? Please clarify in the text.

We believe inferred spike trains provide better resolution and make it easier to compare with quantitative values from electrical recordings. Notice that NC values computed using dF/F can be much larger than those computed by inferred spike trains. For example, see Smith & Hausser 2010 Nat Neurosci. Supplementary Figure S8.

• The sentence seems incomplete or unclear: "That is, there are more high NC pairs that are in-group." Explicit vs what?

We have revised this sentence.

• Figure 1E is unclear to me. What is being plotted? Please add a color bar with the metric and the units for the matrix (left) and in the tuning curves (right panels). If the Y and X axes represent the different classes from the GMM, why are there more than 65 rows? Why is the matrix not full?

We have revised this figure. Fig. 1D is the full 65 x 65 matrix. Fig. 1F has small 3x3 matrices mapping the responses to different TF and SF of gratings. We hope the new version is clearer.

• How are receptive fields defined? How are their long and short axes calculated? How are their limits defined when calculating RF overlap?

We have added further details in the Methods section entitled *“Receptive field analysis”.*